# COMMUNICATION-EFFICIENT FEDERATED NON-LINEAR BANDIT OPTIMIZATION

**Chuanhao Li**[⋆∗] **Chong Liu**[†∗] **Yu-Xiang Wang**[‡]
[⋆]Yale University  [†]University of Chicago  [‡]University of California, Santa Barbara
chuanhao.li.cl2637@yale.edu chongl@uchicago.edu yuxiangw@cs.ucsb.edu

## ABSTRACT

Federated optimization studies the problem of collaborative function optimization among multiple clients (e.g. mobile devices or organizations) under the coordination of a central server. Since the data is collected separately by each client and always remains decentralized, federated optimization preserves data privacy and allows for large-scale computing, which makes it a promising decentralized machine learning paradigm. Though it is often deployed for tasks that are online in nature, e.g., next-word prediction on keyboard apps, most works formulate it as an offline problem. The few exceptions that consider federated bandit optimization are limited to very simplistic function classes, e.g., linear, generalized linear, or non-parametric function class with bounded RKHS norm, which severely hinders its practical usage. In this paper, we propose a new algorithm, named Fed-GO-UCB, for federated bandit optimization with generic non-linear objective function. Under some mild conditions, we rigorously prove that Fed-GO-UCB is able to achieve sub-linear rate for both cumulative regret and communication cost. At the heart of our theoretical analysis are distributed regression oracle and individual confidence set construction, which can be of independent interests. Empirical evaluations also demonstrate the effectiveness of the proposed algorithm.

## 1 INTRODUCTION

Federated optimization is a machine learning method that enables collaborative model estimation over decentralized dataset without data sharing (McMahan et al., 2017; Kairouz et al., 2019). It allows the creation of a shared global model with personal data remaining in local sites instead of being transferred to a central location, and thus reduces the risks of personal data breaches. While the main focus of the state-of-the-art federated optimization is on the offline setting, where the objective is to obtain a good model estimation based on fixed dataset (Li et al., 2019a; Mitra et al., 2021), several recent research efforts have been made to extend federated optimization to the online setting, i.e., federated bandit optimization (Wang et al., 2020; Li & Wang, 2022b; Li et al., 2022a).

Compared with its offline counterpart, federated bandit optimization is characterized by its online interactions with the environment, which continuously provides new data points to the clients over time. The objective of the clients is to collaboratively minimize cumulative regret, which measures how fast they can find the optimal decision, as well as the quality of decisions made during the trial-and-error learning process. This new paradigm greatly improves sample efficiency, as the clients not only collaborate on model estimation, but also actively select informative data points to evaluate in a coordinated manner. Moreover, compared with the standard Bayesian optimization approach (Shahriari et al., 2015), the improved data protection of federated bandit optimization makes it a better choice for applications involving sensitive data, such as recommender systems (Li et al., 2010), clinical trials (Villar et al., 2015) and sequential portfolio selection (Shen et al., 2015). For example, medical data such as disease symptoms and medical reports are very sensitive and private, and are typically stored in isolated medical centers and hospitals (Yang et al., 2019). Federated bandit optimization offers a principled way for different medical institutions to jointly solve optimization problems for smart healthcare applications, while ensuring privacy and communication efficiency.

---

[∗]Equal Contribution

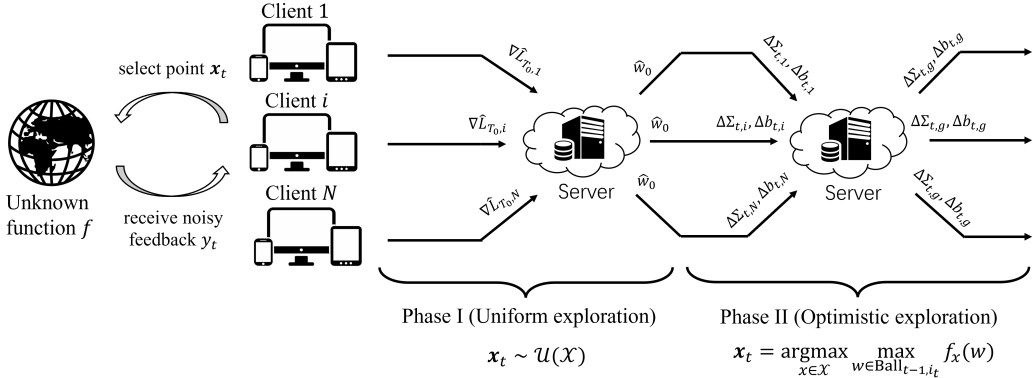

Figure 1: Illustration of Fed-GO-UCB algorithm, which consists of two phases: in Phase I, all clients do uniform exploration for a total number of $T_0$ time steps, and then use the collected data to jointly construct an model $\hat{w}_0$ via iterative distributed optimization; in Phase II, each client constructs a local confidence set for the unknown non-linear function $f$ using gradients w.r.t. the shared model $\hat{w}_0$, based on which they do optimistic exploration. Synchronization of their local statistics happens when a communication event is triggered, which enables coordinated exploration among the clients.

However, despite these potential benefits and the compelling theoretical guarantees, prior works in this direction are limited to very restrictive function classes, e.g., linear (Wang et al., 2020; Li & Wang, 2022a; He et al., 2022), generalized linear (Li & Wang, 2022b), and non-parametric function class with bounded RKHS norm (Li et al., 2022a; 2023; Du et al., 2021), which limits their potential in practical scenarios that typically require more powerful tools in nonlinear modeling, e.g. neural networks. The main challenges in bridging this gap come from two aspects. First, different from offline federated optimization, federated bandit optimization needs to efficiently explore the decision space by actively picking data points to evaluate. This requires a careful construction of confidence sets for the unknown optimal model parameter (Abbasi-yadkori et al., 2011), which is challenging for generic nonlinear functions. Second, for clients to collaboratively estimate confidence sets, occasional communications are required to aggregate their local learning parameters as new data points are collected over time. Prior works consider simple function classes (Wang et al., 2020; Li & Wang, 2022a), so efficient communication can be realized by directly aggregating local sufficient statistics for the closed-form model estimation. However, generic non-linear function places a much higher burden on the communication cost, as iterative optimization procedure is required.

To address these challenges, we propose the Federated Global Optimization with Upper Confidence Bound (Fed-GO-UCB) algorithm, as illustrated in Figure 1. Specifically, Fed-GO-UCB has two phases: Phase I does uniform exploration to sufficiently explore the unknown function; and Phase II does optimistic exploration to let $N$ clients jointly optimize the function. All clients separately choose which points to evaluate, and only share statistics summarizing their local data points with the central server. Details of Fed-GO-UCB are presented in Section 4.1.

**Technical novelties.** Our core technique to address the aforementioned challenges is a novel confidence sets construction that works for generic nonlinear functions, and more importantly, can be updated communication efficiently during federated bandit optimization. Our construction is motivated by Liu & Wang (2023), with non-trivial extensions tailored to federated bandit setting. Specifically, the statistics used for function approximation in Liu & Wang (2023) is computed based on a single sequence of continuously updated models, but in federated bandits, each client has a different sequence of locally updated models. Direct aggregation of such local statistics does not necessarily lead to valid confidence sets. Instead, we propose a new approximation procedure, such that all statistics are computed based on the same fixed model shared by all clients, which is denoted by $\hat{w}_0$. With improved analysis, we show that valid confidence sets can still be constructed. More importantly, this allows direct aggregation of statistics from different clients, so that communication strategies proposed in linear settings can be utilized to reduce frequency of communications. Over the entire horizon, only the estimation of $\hat{w}_0$ at the end of Phase I requires iterative optimization. To further control the communication cost it incurs, we show that $\hat{w}_0$ only needs to be an $O(1/\sqrt{NT})$-

approximation to the empirical risk minimizer that is required in the analysis of Liu & Wang (2023), and adopt a distributed implementation of Gradient Langevin Dynamics (GLD) for its optimization.

**Contributions.** Our main contributions can be summarized as follows.

- To the best of our knowledge, this is the first federated bandit algorithm for generic non-linear function optimization with provable communication efficiency, making it deployment-efficient.
- Under realizable assumption and some other mild conditions, we prove that cumulative regret of Fed-GO-UCB is $\tilde{O}(\sqrt{NT})$ and its communication cost is $\tilde{O}(N^{1.5}\sqrt{T})$.
- Our empirical evaluations show Fed-GO-UCB outperforms existing federated bandit algorithms, which demonstrates the effectiveness of generic non-linear function optimization,

## 2 RELATED WORK

**Centralized global optimization** Most work on global optimization studies the centralized setting where all data points are available on a single machine. Its applications include hyperparameter tuning for deep neural networks (Hazan et al., 2018; Kandasamy et al., 2020) and materials design (Nakamura et al., 2017; Frazier & Wang, 2016). The most popular approach to this problem is Bayesian optimization (BO) (Shahriari et al., 2015; Frazier, 2018), which is closely related to bandit problems (Li et al., 2019b; Foster & Rakhlin, 2020). BO typically assumes the unknown objective function is drawn from some Gaussian Processes (GP). The learner sequentially choose points to evaluate and then improve its estimation via posterior update. Classical BO algorithms include GP-UCB (Srinivas et al., 2010), GP-EI (Jones et al., 1998), and GP-PI (Kushner, 1964). To improve heterogeneous modeling of the objective function and mitigate over-exploration, Trust region BO (Eriksson et al., 2019) that uses multiple local optimization runs is proposed. In this line of research, the closes work to ours is Liu & Wang (2023), which also considers global approximation of generic nonlinear functions, though it's not suitable for federated setting as discussed in Section 1.

**Federated bandit optimization** Another closely related line of research is federated/distributed bandits, where multiple agents collaborate in pure exploration (Hillel et al., 2013; Tao et al., 2019; Du et al., 2021), or regret minimization (Wang et al., 2020; Li & Wang, 2022a;b). However, most of these works make linear model assumptions, and thus the clients can efficiently collaborate by transferring the $O(d_x^2)$ sufficient statistics for closed-form model estimation, where $d_x$ is input data dimension. The closest works to ours are Wang et al. (2020); Dubey & Pentland (2020); Li & Wang (2022a;b), which uses event-triggered communication strategies to obtain sub-linear communication cost over time, i.e., communication only happens when sufficient amount of new data has been collected. There is also recent work by Dai et al. (2022) that studies federated bandits with neural function approximation, but it still relies on GP with a Neural Tangent Kernel in their analysis, which is intrinsically linear. More importantly, this analysis assumes the width of the neural network is much larger than the number of samples, while our results do not require such over-parameterization.

## 3 PRELIMINARIES

We consider the problem of finding a global maximum solution to an unknown non-linear black-box function $f$, i.e.,

$$\mathbf{x}^* = \arg\max_{\mathbf{x} \in \mathcal{X}} f(\mathbf{x}).$$

Different from previous works, we consider a decentralized system of 1) $N$ clients that selects data points to evaluate, and 2) a central server that coordinates the communication among the clients. The clients cannot directly communicate with each other, but only with the central server, i.e., a star-shaped communication network as shown in Figure 1. In each round, $N$ clients interact with the unknown function $f$ in a round-robin manner, for a total number of $T$ rounds, so the total number of interactions is $NT$. Let $[N]$ denote the integer set $\{1, 2, ..., N\}$. Specifically, at round $l \in [T]$, each client $i \in [N]$ selects a point $\mathbf{x}_t$ from the set $\mathcal{X}$, and has a zeroth-order noisy function observation:

$$y_t = f(\mathbf{x}_t) + \eta_t \in \mathbb{R}, \tag{1}$$

where the subscript $t := N(l-1) + i$ indicates this is the $t$-th interaction between the system and the function $f$, i.e., the $t$-th time function $f$ is evaluated at a selected point $\mathbf{x}_t$, and $\eta_t$ is independent, zero-mean, $\sigma$-sub-Gaussian noise, for $t \in [NT]$.

We adopt the classical definition of cumulative regret to evaluate the algorithm performance. It is defined as $R_{NT} = \sum_{t=1}^{NT} f(\mathbf{x}^*) - f(\mathbf{x}_t)$ for $NT$ interactions. Following Wang et al. (2020), we define the communication cost $C_{NT}$ as the total amount of real numbers being transferred across the system during the $NT$ interactions with function $f$.

Let $\mathcal{U}$ denote uniform distribution. W.l.o.g., let $\mathcal{X} \subseteq [0,1]^{d_x}$ and $\mathcal{Y} \subset \mathbb{R}$ denote the domain and range of unknown function $f$. We are working with a parametric function class $\mathcal{F} := \{f_w : \mathcal{X} \to \mathcal{Y} | w \in \mathcal{W}\}$ to approximate $f$ where the parametric function class is controlled by the parameter space $\mathcal{W}$. For a parametric function $f_w(x)$, let $\nabla f_w(x)$ denote the gradient taken w.r.t. $x$ and $\nabla f_x(w)$ denote the gradient taken w.r.t. $w$. We use $i_t \in [N]$ as the index of the client that evaluates point $x_t$ at time step $t$. We denote the sequence of time steps corresponding to data points that have been evaluated by client $i$ up to time $t$ as $D_{t,i}^l = \{1 \leq s \leq t : i_s = i\}$ for $t = 1, 2, \ldots, NT$. In addition, we denote the sequence of time steps corresponding to the data points that have been used to update client $i$'s local model as $D_{t,i}$, which include both data points collected locally and those received from the server. If client $i$ never receives any communication from the server, $D_{t,i} = D_{t,i}^l$; otherwise, $D_{t,i}^l \subset D_{t,i} \subseteq [t]$. Moreover, when each new data point evaluated by any client in the system is readily communicated to all the other clients, we recover the centralized setting, i.e., $D_{t,i} = [t], \forall i, t$. For completeness, we list all notations in Appendix A.

Here we list all assumptions that we use throughout this paper. The first two assumptions are pretty standard and the last assumption comes from previous works.

**Assumption 1** (Realizabilty). *There exists $w^\star \in \mathcal{W}$ such that the unknown objective function $f = f_{w^\star}$. Also, assume $\mathcal{W} \subset [0,1]^{d_w}$. This is w.l.o.g. for any compact $\mathcal{W}$.*

**Assumption 2** (Bounded, differentiable and smooth function approximation). *There exist constants $F, C_g, C_h > 0$, s.t. $|f_x(w)| \leq F$, $\|\nabla f_x(w)\|_2 \leq C_g$, and $\|\nabla^2 f_x(w)\|_{\text{op}} \leq C_h$, $\forall x \in \mathcal{X}, w \in \mathcal{W}$.*

**Assumption 3** (Geometric conditions on the loss function (Liu & Wang, 2023; Xu et al., 2018)). *$L(w) = \mathbb{E}_{x \sim \mathcal{U}}(f_x(w) - f_x(w^\star))^2$ satisfies $(\tau, \gamma)$-growth condition or local $\mu$-strong convexity at $w^\star$, i.e., $\forall w \in \mathcal{W}$,*

$$\min\left\{\frac{\mu}{2}\|w - w^\star\|_2^2, \frac{\tau}{2}\|w - w^\star\|_2^\gamma\right\} \leq L(w) - L(w^\star),$$

*for constants $\mu, \tau > 0, \mu < d_w, 0 < \gamma < 2$. $L(w)$ satisfies dissipative assumption, i.e., $\nabla L(w)^\top w \geq C_i \|w\|_2^2 - C_j$ for some constants $C_i, C_j > 0$, and a $c$-local self-concordance assumption at $w^\star$, i.e., $(1-c)^2 \nabla^2 L(w^\star) \preceq \nabla^2 L(w^\star) \preceq (1-c)^{-2} \nabla^2 L(w^\star)$, $\forall w$ s.t. $\|w - w^\star\|_{\nabla^2 L(w)} \leq c$.*

Assumption 2 implies there exists a constant $\zeta > 0$, such that $\|\nabla^2 L(w^\star)\|_{op} \leq \zeta$. For example, it suffices to take $\zeta = 2C_g^2$. Assumption 3 is made on the expected loss function $L$ w.r.t. parameter $w$ rather than function $f$, and is strictly weaker than strong convexity as it only requires strong convexity in the small neighboring region around $w^*$. For $w$ far away from $w^*$, $L(w)$ can be highly non-convex since only growth condition needs to be satisfied. The dissipative assumption is typical for stochastic differential equation and diffusion approximation (Raginsky et al., 2017; Zhang et al., 2017). In our paper, it is only needed for the convergence analysis of Algorithm 2, and may be relaxed by adopting other non-convex optimization methods with global convergence guarantee.

## 4 METHODOLOGY

### 4.1 FED-GO-UCB ALGORITHM

In this paper, we develop an algorithm that allows Bayesian optimization style active queries to work for general supervised learning-based function approximation under federated optimization scheme.

**Phase I** In Step 1 of Phase I, the algorithm evaluates the unknown function $f$ at uniformly sampled points for a total number of $T_0$ times, where $T_0$ is chosen to be large enough such that function $f$ can be sufficiently explored. By the end of time step $T_0$, each client $i \in [N]$ has collected a local dataset $\{(\mathbf{x}_s, y_s)\}_{s \in D_{T_0,i}^l}$ (by definition $\cup_{i \in [N]} D_{T_0,i}^l = [T_0]$). In Step 2, we call the distributed regression oracle, denoted as *Oracle*, to jointly learn model parameter $\hat{w}_0$, by optimizing equation 2 below.

$$\min_{w \in \mathcal{W}} \hat{L}_{T_0}(w) := \frac{1}{T_0} \sum_{i=1}^N \hat{L}_{T_0,i}(w), \tag{2}$$

---

**Algorithm 1** Fed-GO-UCB

---

**Input:** Phase I length $T_0$, Time horizon (Phase II length) $NT$, *Oracle*, number of iterations $n$ to execute *Oracle*, communication threshold $\gamma$, regularization weight $\lambda$.

**Phase I** (Uniform exploration)

1: **for** $t = 1, 2, \ldots, T_0$ **do** client $i_t \in [N]$ evaluates point $x_t \sim \mathcal{U}(\mathcal{X})$, and observe $y_t$
2: Execute *Oracle* (e.g., Algorithm 2) over $N$ clients to obtain $\hat{w}_0$

**Phase II** (Optimistic exploration)

1: Initialize $\hat{w}_{T_0,i} = \hat{w}_0$, $\Sigma_{T_0,i} = \lambda \mathbf{I}$, $b_{T_0,i} = \mathbf{0}$, $D_{T_0,i} = \emptyset$, for each client $i \in [N]$
2: **for** $t = 1, \ldots, NT$ **do**
3:     Client $i_t$ evaluates point $x_t = \arg\max_{x \in \mathcal{X}} \max_{w \in \mathrm{Ball}_{t-1,i_t}} f_x(w)$ and observe $y_t$
4:     Client $i_t$ updates $\Sigma_{t,i_t} = \Sigma_{t-1,i_t} + \nabla f_{x_t}(\hat{w}_0) \nabla f_{x_t}(\hat{w}_0)^\top$, $b_{t,i_t} = b_{t-1,i_t} + \nabla f_{x_t}(\hat{w}_0)(\nabla f_{x_t}(\hat{w}_0)^\top \hat{w}_0 + y_t - f_{x_t})$, and $\Delta\Sigma_{t,i_t} = \Delta\Sigma_{t-1,i_t} + \nabla f_{x_t}(\hat{w}_0) \nabla f_{x_t}(\hat{w}_0)^\top$, $\Delta b_{t,i_t} = \Delta b_{t-1,i_t} + \nabla f_{x_t}(\hat{w}_0) \nabla f_{x_t}(\hat{w}_0)^\top \hat{w}_0 + y_t - f_{x_t}$
5:     Client $i_t$ updates $\hat{w}_{t,i_t}$ by equation 5 and $\mathrm{Ball}_{t,i_t}$ by equation 4; sets index set $D_{t,i_t} = D_{t-1,i_t} \cup \{t\}$
    *# Check whether global synchronization is triggered*
6:     **if** $\left(|\mathcal{D}_{t,i_t}| - |\mathcal{D}_{t_{\mathrm{last}},i_t}|\right) \log \frac{\det(\Sigma_{t,i_t})}{\det(\Sigma_{t,i_t} - \Delta\Sigma_{t,i_t})} > \gamma$ **then**
7:         All clients upload $\{\Delta\Sigma_{t,i}, \Delta b_{t,i}\}$, and reset $\Delta\Sigma_{t,i} = \mathbf{0}$, $\Delta b_{t,i} = \mathbf{0}$ for $i = 1, \ldots, N$
8:         Server aggregates $\Sigma_{t,g} = \Sigma_{t_{\mathrm{last}},g} + \sum_{i=1}^N \Delta\Sigma_{t,i}$, $b_{t,g} = b_{t_{\mathrm{last}},g} + \sum_{i=1}^N \Delta b_{t,i}$
9:         All clients download $\{\Sigma_{t,g}, b_{t,g}\}$, and update $\Sigma_{t,i} = \Sigma_{t,g}, b_{t,i} = b_{t,g}$ for $i = 1, \ldots, N$
10:        Set $t_{\mathrm{last}} = t$

**Output:** $\hat{x} \sim \mathcal{U}(\{x_1, \ldots, x_T\})$.

---

where $\hat{L}_{T_0,i}(w) = \sum_{s \in D_{T_0,i}^l} \left[ y_s - f_w(\mathbf{x}_s^\top) \right]^2$ is the squared error loss on client $i$'s local dataset. It is worth noting that, executing *Oracle* to compute the exact minimizer $\hat{w}_0^\star := \arg\min_{w \in \mathcal{W}} \hat{L}_{T_0}(w)$ as in the prior work by Liu & Wang (2023) is unreasonable in our case, since it requires infinite number of iterations, which leads to infinite communication cost. Instead, we need to relax the requirement by allowing for an approximation error $\epsilon$, such that *Oracle* only need to output $\hat{w}_0$ that satisfies

$$|\hat{L}_{T_0}(\hat{w}_0) - \hat{L}_{T_0}(\hat{w}_0^\star)| \leq \epsilon. \tag{3}$$

*Oracle* can be any distributed non-convex optimization method with global convergence guarantee, i.e., we can upper bound the number of iterations required, denoted as $n$, to attain $\epsilon$, for some $\epsilon \geq 0$.

**Remark 4.** *In this paper we adopt Gradient Langevin Dynamics (GLD) based methods to optimize equation 2, which are known to have global convergence guarantee to the minimizer of non-convex objectives under smooth and dissipative assumption (Assumption 2 and Assumption 3). GLD based methods work by introducing a properly scaled isotropic Gaussian noise to the gradient descent update at each iteration, which allows them to escape local minima. Specifically, we use a distributed implementation of GLD, which is given in Algorithm 2. It requires $n = O(\frac{d_x}{\epsilon\nu} \cdot \log(\frac{1}{\epsilon}))$ iterations to attain $\epsilon$ approximation (with step size set to $\tau_1 \leq \epsilon$), where $\nu = O(e^{-\tilde{O}(d_x)})$ denotes the spectral gap of the discrete-time Markov chain generated by GLD (Theorem 3.3 of Xu et al. (2018)).*

---

**Algorithm 2**    Distributed-GLD-Update

---

1: **Input:** total iterations $n$; step size $\tau_1 > 0$; inverse temperature parameter $\tau_2 > 0$; $w^{(0)} = \mathbf{0}$
2: **for** $k = 0, 1, \ldots, n-1$ **do**
3:     Server sends $w^{(k)}$ to all clients, and receives local gradients $\nabla \hat{L}_{T_0,i}(w^{(k)})$ for $i \in [N]$ back
4:     Server aggregates local gradients $\nabla \hat{L}_{T_0}(w^{(k)}) = \frac{1}{T_0} \sum_{i=1}^N \nabla \hat{L}_{T_0,i}(w^{(k)})$
5:     Server randomly draws $z_k \sim \mathcal{N}(\mathbf{0}, \mathbf{I}_{d \times d})$
6:     Server computes update $w^{(k+1)} = w^{(k)} - \tau_1 \nabla \hat{L}_{T_0}(w^{(k)}) + \sqrt{2\tau_1/\tau_2} z_k$
7: **Output:** $\hat{w}_0 = w^{(K)}$

---

**Phase II**    At the beginning of Phase II, the estimator $\hat{w}_0$ obtained in Phase I is sent to each client, which will be used to construct the confidence sets about the unknown parameter $w^\star$. Specifically,

at each time step $t = 1, 2, \ldots, NT$, the confidence set constructed by client $i_t$ is a ball defined as

$$\text{Ball}_{t,i_t} := \{w : \|w - \hat{w}_{t,i_t}\|_{\Sigma_{t,i_t}}^2 \leq \beta_{t,i_t}\}, \tag{4}$$

such that with the choice of $\beta_{t,i_t}$ in Lemma 8, $w^\star \in \text{Ball}_{t,i_t}, \forall t$ with probability at least $1 - \delta$. The center of this ball is defined as

$$\hat{w}_{t,i_t} := \Sigma_{t,i_t}^{-1} b_{t,i_t} + \lambda \Sigma_{t,i_t}^{-1} \hat{w}_0, \tag{5}$$

where the matrix

$$\Sigma_{t,i_t} := \lambda \mathbf{I} + \sum_{s \in D_{t,i_t}} \nabla f_{\mathbf{x}_s}(\hat{w}_0) \nabla f_{\mathbf{x}_s}(\hat{w}_0)^\top, \tag{6}$$

and vector $b_{t,i_t} := \sum_{s \in D_{t,i_t}} \nabla f_{\mathbf{x}_s}(\hat{w}_0) [\nabla f_{\mathbf{x}_s}(\hat{w}_0)^\top \hat{w}_0 + y_s - f_{\mathbf{x}_s}(\hat{w}_0)]$. Note $\nabla f_{\mathbf{x}_s}(\hat{w}_0)$ is the gradient of our parametric function taken w.r.t parameter $\hat{w}_0$, rather than the gradient of the unknown objective function. The statistics $\Sigma_{t,i}, b_{t,i}$ for all client $i$ and time $t$ are constructed using gradients w.r.t. the same model $\hat{w}_0$. This is essential in federated bandit optimization, as it allows the clients to jointly construct the confidence set by directly aggregating their local updates, denoted by $\Delta\Sigma_{t,i}, \Delta b_{t,i}$. In comparison, although the statistics used to construct the confidence sets in Liu & Wang (2023) are computed based on different models and lead to tigher results, they impede collaboration among clients and cannot be directly used in our case.

In Phase II, exploration is conducted following "Optimism in the Face of Uncertainty (OFU)" principle, i.e., at time step $t$, client $i_t$ selects point $\mathbf{x}_t \in \mathcal{X}$ to evaluate via joint optimization over $x \in \mathcal{X}$ and $w \in \text{Ball}_{t-1,i_t}$, as shown in line 3. The newly obtained data point $(\mathbf{x}_t, y_t)$ will then be used to update client $i_t$'s confidence set as shown in line 4-5. In order to ensure communication efficiency during the collaborative global optimization across $N$ clients, an event-triggered communication protocol is adopted, as shown in line 6. Intuitively, this event measures the amount of new information collected by client $i_t$ since last global synchronization. If the value of LHS exceeds threshold $\gamma$, another global synchronization will be triggered, such that the confidence sets of all $N$ clients will be synchronized as shown in line 7-9. In Section 5, we will show that with proper choice of $\gamma$, the total number of synchronizations can be reduced to $\tilde{O}(\sqrt{N})$, without degrading the performance.

## 4.2 THEORETICAL RESULTS

**Theorem 5** (Cumulative regret and communication cost of Fed-GO-UCB). *Suppose Assumption 1, 2, and 3 hold. Let $C$ denote a universal constant and $C_\lambda$ denote a constant that is independent to $T$. Under the condition that $T \geq \frac{C d_w^2 F^4 \iota^2}{N} \cdot \max\{\frac{\mu^{\gamma/(2-\gamma)}}{\tau^{2/(2-\gamma)}}, \frac{\zeta}{\mu c^2}\}^2$, where $\iota$ is the logarithmic term depending on $T_0$, $C_h$, and $2/\delta$. Algorithm 1 with parameters $T_0 = \sqrt{NT}$, $\lambda = C_\lambda \sqrt{NT}$, $\gamma = \frac{d_w F^4 T}{\mu^2 N}$, and $\epsilon \leq \frac{8}{C\sqrt{NT}}$, has cumulative regret*

$$R_{\text{phase I + phase II}} = \tilde{O}\left(\sqrt{NT}F + \sqrt{NT}d_w^3 F^4/\mu^2 + N d_w^4 F^4/\mu^2\right),$$

*with probability at least $1 - \delta$, and communication cost*

$$C_{\text{phase I + phase II}} = \tilde{O}(N^{1.5}\sqrt{T}d_w d_x \exp(d_x) + N^{1.5}d_w^2 \mu/F^2).$$

Theorem 5 shows that our proposed Fed-GO-UCB algorithm matches the regret upper bound of its centralized counterpart, GO-UCB algorithm by Liu & Wang (2023), while only requiring communication cost that is sub-linear in $T$. We should note that the $O(\sqrt{T})$ dependence in communication cost is due to the iterative optimization procedure to compute $\hat{w}_0$ at the end of Phase I, which also exists for prior works in federated bandits that requires iterative optimization (Li & Wang, 2022b).

## 5 PROOF OVERVIEW

In this section, to highlight our technical contributions, we provide a proof sketch of the theoretical results about cumulative regret and communication cost that are presented in Theorem 5. All auxiliary lemmas are given in Appendix C and complete proofs are presented in Appendix D.

## 5.1 PHASE I. UNIFORM EXPLORATION & DISTRIBUTED REGRESSION ORACLE

**Cumulative Regret and Communication Cost in Phase I**  Recall from Section 4.1 that, $N$ clients conduct uniform exploration in Phase I, which constitutes a total number of $T_0$ interactions with the environment. By Assumption 2, we know that the instantaneous regret has a uniform upper bound $r_t := f(\mathbf{x}^\star) - f(\mathbf{x}_t) \leq 2F$, so for a total number of $T_0$ time steps, the cumulative regret incurred in Phase I, denoted as $R_{\text{phase I}}$, can be upper bounded by $R_{\text{phase I}} = \sum_{t=1}^{T_0} r_t \leq 2T_0 F$. Choice of $T_0$ value will be discussed in Section 5.2, as it controls the quality of $\hat{w}_0$, which further affects the constructed confidence sets used for optimistic exploration.

Moreover, the only communication cost in Phase I is incurred when executing *Oracle*, i.e., the distributed regression oracle, for $n$ iterations. In each iteration, $N$ clients need to upload their local gradients to the server, and then receive the updated global model back (both with dimension $d_w$). Therefore, the communication cost incurred during Phase I is $C_{\text{phase I}} = 2nNd_w$.

**Distributed Regression Oracle Guarantee**  At the end of Phase I, we obtain an estimate $\hat{w}_0$ by optimizing equation 2 using *Oracle*. As we mentioned in Section 4.2, $\hat{w}_0$ will be used to construct the confidence sets, and thus to prepare for our analysis of the cumulative regret in Phase II, we establish the following regression oracle lemmas.

**Lemma 6.** *There is an absolute constant $C'$, such that after time step $T_0$ in Phase I of Algorithm 1 and under the condition that approximation error $\epsilon \leq 1/(C'T_0)$, with probability at least $1 - \delta/2$, regression oracle estimated $\hat{w}_0$ satisfies $\mathbb{E}_{x \sim \mathcal{U}}[(f_x(\hat{w}_0) - f_x(w^\star))^2] \leq C'd_wF^2\iota/T_0$, where $\iota$ is the logarithmic term depending on $T_0, C_h, 2/\delta$.*

Lemma 6 is adapted from Lemma 5.1 of Liu & Wang (2023) to account for the additional approximation error from the distributed regression oracle. Specifically, instead of proving the risk bound for the exact minimizer $\hat{w}_0^\star$, we consider $\hat{w}_0$ that satisfies $|\hat{L}_{T_0}(\hat{w}_0) - \hat{L}_{T_0}(\hat{w}_0^\star)| \leq \epsilon$ for some constant $\epsilon$. As discussed in Section 4.1, this relaxation is essential for the communication efficiency in federated bandit optimization. And Lemma 6 shows that, by ensuring $\epsilon \leq 1/(C'T_0)$, we can obtain the same regression oracle guarantee as in the centralized setting (Liu & Wang, 2023). As discussed in Remark 4, this condition can be satisfied with $n = O(\frac{C'd_xT_0}{\nu} \cdot \log(C'T_0))$ number of iterations. With Lemma 6, we can establish Lemma 7 below using the same arguments as Liu & Wang (2023).

**Lemma 7** (Regression oracle guarantee (Theorem 5.2 of Liu & Wang (2023)))**.** *Under Assumption 1, 2, and 3, and by setting $\epsilon \leq 1/(C'T_0)$, there exists an absolute constant $C$ such that after time step $T_0$ in Phase I of Algorithm 1, where $T_0$ satisfies $T_0 \geq Cd_wF^2\iota \cdot \max\left\{\frac{\mu^{\gamma/(2-\gamma)}}{\tau^{2/(2-\gamma)}}, \frac{\zeta}{\mu c^2}\right\}$, with probability at least $1 - \delta/2$, regression oracle estimated $\hat{w}_0$ satisfies $\|\hat{w}_0 - w^\star\|_2^2 \leq \frac{Cd_wF^2\iota}{\mu T_0}$.*

## 5.2 PHASE II. CONFIDENCE SET CONSTRUCTION & OPTIMISTIC EXPLORATION

**Confidence Set Construction**  In Phase II of Algorithm 1, each client $i$ selects its next point to evaluate based on OFU principle, which requires construction of the confidence set in equation 4. The following lemma specifies the proper choice of $\beta_{t,i_t}$, such that $\text{Ball}_{t,i_t}$ contains true parameter $w^\star$ for all $t \in [T]$ with high probability.

**Lemma 8.** *Under Assumption 1, 2, & 3 and by setting $\beta_{t,i_t} = \tilde{\Theta}\left(d_w\sigma^2 + d_wF^2/\mu + d_w^3F^4/\mu^2\right)$, $T_0 = \sqrt{NT}$, and $\lambda = C_\lambda\sqrt{NT}$, then $\|\hat{w}_{t,i_t} - w^*\|_{\Sigma_{t,i_t}}^2 \leq \beta_{t,i_t}$, with probability at least $1 - \delta$, $\forall t \in [NT]$ in Phase II of Algorithm 1.*

**Cumulative Regret in Phase II**  Thanks to the confidence sets established in Lemma 8, Algorithm 1 can utlize a communication protocol similar to the ones designed for federated linear bandits (Wang et al., 2020; Dubey & Pentland, 2020) during Phase II, while providing much diverse modeling choices. Therefore, our analysis of the communication regret and communication cost incurred in Phase II follows a similar procedure as its linear counterparts.

Denote the total number of global synchronizations (total number of times the event in line 6 of Algorithm 1 is true) over time horizon $T$ as $P \in [0, NP]$. Then we use $t_p$ for $p \in [P]$ to denote the time step when the $p$-th synchronization happens (define $t_0 = 0$), and refer to the sequence of time steps in-between two consecutive synchronizations as an epoch, i.e., the $p$-th epoch is $[t_{p-1} + 1, t_p]$.

Similar to Wang et al. (2020); Dubey & Pentland (2020); Li & Wang (2022b), for the cumulative regret analysis in Phase II, we decompose $P$ epochs into good and bad epochs, and then analyze them separately. Specifically, consider an imaginary centralized agent that has immediate access to each data point in the learning system, and we let this centralized agent executes the same model update rule and arm selection rule as in line 3-5 of Algorithm 1. Then the covariance matrix maintained by this agent can be defined as $\Sigma_t = \sum_{s=1}^{t} \nabla f_{x_s}(\hat{w}_0) \nabla f_{x_s}(\hat{w}_0)^\top$ for $t \in [NT]$. Then the $p$-th epoch is called a good epoch if $\ln\left(\frac{\det(\Sigma_{t_p})}{\det(\Sigma_{t_{p-1}})}\right) \leq 1$, otherwise it is a bad epoch. Note that based on Lemma 9, we have $\ln(\det(\Sigma_{NT})/\det(\lambda I)) \leq d_w \log\left(1 + \frac{NTC_g^2}{d_w \lambda}\right) := R$. Since $\ln(\frac{\det(\Sigma_{t_1})}{\det(\lambda I)}) + \ln(\frac{\det(\Sigma_{t_2})}{\det(\Sigma_{t_1})}) + \cdots + \ln(\frac{\det(\Sigma_{NT})}{\det(\Sigma_{t_B})}) = \ln(\frac{\det(\Sigma_{NT})}{\det(\lambda I)}) \leq R$, and due to the pigeonhole principle, there can be at most $R$ bad epochs. Then with standard optimistic argument (Abbasi-yadkori et al., 2011), we can show that the cumulative regret incurred in good epochs $R_{\text{good}} = \tilde{O}(\frac{d_w^3 F^4 \sqrt{NT}}{\mu^2})$, which matches the regret of centralized algorithm by (Liu & Wang, 2023). Moreover, by design of the event-triggered communication in line 6 of Algorithm 1, we can show that the cumulative regret incurred in any bad epoch $p$ can be bounded by $\sum_{t=t_{p-1}+1}^{t_p} r_t \leq N\sqrt{16\beta_{NT}\gamma + 8\beta_{NT}^2 C_h^2/C_\lambda^2}$, where $\beta_{NT} = O(\frac{d_w^3 F^4}{\mu^2})$ according to Lemma 8. Since there can be at most $R$ bad epochs, the cumulative regret incurred in bad epochs $R_{\text{bad}} = \tilde{O}(Nd_w\sqrt{\frac{d_w^3 F^4}{\mu^2}}\sqrt{\gamma} + Nd_w\frac{d_w^3 F^4}{\mu^2})$. By setting communication threshold $\gamma = \frac{d_w F^4 T}{\mu^2 N}$, we have $R_{\text{bad}} = \tilde{O}(\frac{d_w^3 F^4 \sqrt{NT}}{\mu^2} + \frac{d_w^4 F^4 N}{\mu^2})$. Combining cumulative regret incurred during both good and bad epochs, we have

$$R_{\text{phase II}} = R_{\text{good}} + R_{\text{bad}} = \tilde{O}\left(\sqrt{NT}d_w^3 F^4/\mu^2 + Nd_w^4 F^4/\mu^2\right).$$

**Communication Cost in Phase II** Consider some $\alpha > 0$. By pigeon-hole principle, there can be at most $\lceil\frac{NT}{\alpha}\rceil$ epochs with length (total number of time steps) longer than $\alpha$. Then we consider some epoch with less than $\alpha$ time steps, similarly, we denote the first time step of this epoch as $t_s$ and the last as $t_e$, i.e., $t_e - t_s < \alpha$. Since the users appear in a round-robin manner, the number of interactions for any user $i \in [N]$ satisfies $|D_{t_e,i}| - |D_{t_s,i}| < \frac{\alpha}{N}$. Due to the event-triggered in line 6 of Algorithm 1, we have $\log\frac{\det(\Sigma_{t_e})}{\det(\Sigma_{t_s})} > \frac{\gamma N}{\alpha}$. Using the pigeonhole principle again, we know that the number of epochs with less than $\alpha$ time steps is at most $\lceil\frac{R\alpha}{\gamma N}\rceil$. Therefore, the total number of synchronizations $P \leq \lceil\frac{NT}{\alpha}\rceil + \lceil\frac{R\alpha}{\gamma N}\rceil$, and the RHS can be minimized by choosing $\alpha = N\sqrt{\gamma T/R}$, so that $P \leq 2\sqrt{TR/\gamma}$. With $\gamma = \frac{d_w F^4 T}{\mu^2 N}$, $P \leq 2\sqrt{\frac{N\log(1+NTC_g^2/(d_w\lambda))\mu^2}{F^4}} = \tilde{O}(\sqrt{N\mu^2/F^4})$. At each global synchronization, Algorithm 1 incurs $2N(d_w^2 + d_w)$ communication cost to update the statistics. Therefore, $C_{\text{phase II}} = P \cdot 2N(d_w^2 + d_w) = (N^{1.5}d_w^2\mu/F^2)$.

# 6 EXPERIMENTS

In order to evaluate Fed-GO-UCB's empirical performance and validate our theoretical results in Theorem 5, we conducted experiments on both synthetic and real-world datasets. Due to the space limit, here we only discuss the experiment setup and results on synthetic dataset. More discussions about the experiment setup and results on real-world datasets are presented in Appendix E.

For synthetic dataset, we consider two test functions, $f_1(\mathbf{x}) = -\sum_{i=1}^{4}\bar{\alpha}_i\exp(-\sum_{j=1}^{6}\bar{A}_{ij}(\mathbf{x}_j - \bar{P}_{ij})^2)$ (see values of $\bar{\alpha}, \bar{A}, \bar{P}$ in appendix) and $f_2(\mathbf{x}) = 0.1\sum_{i=1}^{8}\cos(5\pi\mathbf{x}_i) - \sum_{i=1}^{8}\mathbf{x}_i^2$. The decision set $\mathcal{X}$ is finite (with $|\mathcal{X}| = 50$), and is generated by uniformly sampling from $[0,1]^6$ and $[-1,1]^8$, respectively. We choose $\mathcal{F}$ to be a neural network with two linear layers, i.e., the model $\hat{f}(\mathbf{x}) = W_2 \cdot \sigma(W_1\mathbf{x} + c_1) + c_2$, where the parameters $W_1 \in \mathbb{R}^{25,d_x}, c_1 \in \mathbb{R}^{25}, W_2 \in \mathbb{R}^{25}, c_2 \in \mathbb{R}$, and $\sigma(z) = 1/(1+\exp(-z))$. Results (averaged over 10 runs) are reported in Figure 2. We included DisLinUCB (Wang et al., 2020), Fed-GLB-UCB (Li & Wang, 2022b), ApproxDisKernelUCB (Li et al., 2022a), One-GO-UCB, and N-GO-UCB (Liu & Wang, 2023) as baselines. One-GO-UCB learns one model for all clients by immediately synchronizing every data point, and N-GO-UCB learns one model for each client with no communication. From Figure, 2, we can see that Fed-GO-UCB and One-GO-UCB have much smaller regret than other baseline algorithms, demonstrating

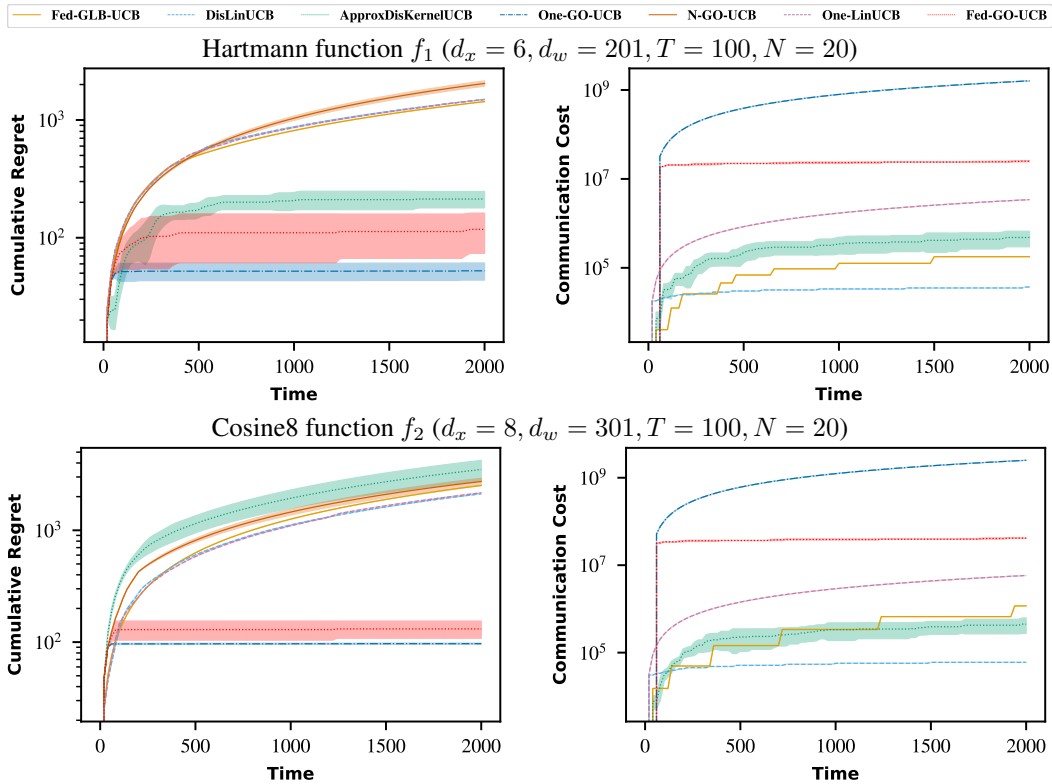

Figure 2: Comparison of cumulative regret and communication cost on synthetic test functions.

the superiority of neural network approximation for the global non-linear optimization, though inevitably this comes at the expense of higher communication cost due to transferring statistics of size $d_w^2 \approx 10^4$ (compared with $d_x^2 \approx 10^2$ for the baselines). Nevertheless, we can see that the communication cost of Fed-GO-UCB is significantly lower than One-GO-UCB and it grows at a sub-linear rate over time, which conforms with our theoretical results in Theorem 5.

## 7  CONCLUSIONS AND FUTURE WORK

Despite the potential of federated optimization in high-impact applications, such as, clinical trial optimization, hyperparameter tuning, and drug discovery, there is a gap between current theoretical studies and practical usages, i.e.,, federated optimization is often needed in online tasks, like next-word prediction on keyboard apps, but most existing works formulate it as an offline problem. To bridge this gap, some recent works propose to study federated bandit optimization problem, but their objective functions are limited to simplistic classes, e.g., linear, generalized linear, or non-parametric function class with bounded RKHS norm, which limits their potential in real-world applications.

In this paper, we propose the first federated bandit optimization method, named Fed-GO-UCB, that works with generic non-linear objective functions. Under some mild conditions, we rigorously prove that Fed-GO-UCB is able to achieve $\tilde{O}(\sqrt{NT})$ cumulative regret and $\tilde{O}(N^{1.5}\sqrt{T} + N^{1.5})$ communication cost where $T$ is time horizon and $N$ is number of clients. Our technical analysis builds upon Xu et al. (2018); Liu & Wang (2023) and the main novelties lie in the distributed regression oracle and individual confidence set construction, which makes collaborative exploration under federated setting possible. In addition, extensive empirical evaluations are performed to validate the effectiveness of our algorithm, especially its ability in approximating nonlinear functions.

One interesting future direction is to generalize our work to heterogeneous clients, i.e., each client $i \in [N]$ has a different reward function $f_i$, that can be assumed to be close to each other as in Dubey & Pentland (2021), or have shared components as in Li & Wang (2022a). This allows us to better model the complex situations in reality, especially when personalized decisions are preferred.

ACKNOWLEDGMENTS

The work was partially supported by NSF Awards #2007117 and #2003257. The work was partially done when Chong Liu was at UCSB and when Chuanhao Li was at University of Virginia. We thank ICLR reviewers and the area chair for their valuable input that led to improvements to the paper.

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

# A    NOTATION TABLE

Table 1: Symbols and notations.

| Symbol | Definition | Description |
|---|---|---|
| $\|A\|_{\mathrm{op}}$ | | operator norm of matrix $A$ |
| $\mathrm{Ball}_t$ | equation 4 | parameter uncertainty region at round $t$ |
| $\beta_t$ | given in Lemma 8 | parameter uncertainty region radius at round $t$ |
| $C, \zeta$ | | constants |
| $d_x$ | | domain dimension |
| $d_w$ | | parameter dimension |
| $\delta$ | | failure probability |
| $\varepsilon$ | | covering number discretization distance |
| $\eta$ | $\sigma$-sub-Gaussian | observation noise |
| $f_w(\mathbf{x})$ | | objective function at $\mathbf{x}$ parameterized by $w$ |
| $f_{\mathbf{x}}(w)$ | | objective function at $w$ parameterized by $\mathbf{x}$ |
| $\nabla f_{\mathbf{x}}(w)$ | | 1st order derivative w.r.t. $w$ parameterized by $\mathbf{x}$ |
| $\nabla^2 f_{\mathbf{x}}(w)$ | | 2nd order derivative w.r.t. $w$ parameterized by $\mathbf{x}$ |
| $F$ | | function range constant bound |
| $\iota, \iota', \iota''$ | | logarithmic terms |
| $L(w)$ | $\mathbb{E}[(f_x(w) - f_x(w^*))^2]$ | expected loss function |
| $\lambda$ | | regularization parameter |
| $\mu$ | | strong convexity parameter |
| $[T]$ | $\{1, 2, ..., T\}$ | integer set of size $T$ |
| $N$ | | number of agents |
| Oracle | | regression oracle |
| $n$ | | number of iterations to execute Oracle |
| $\epsilon$ | | approximation error guaranteed of Oracle |
| $\gamma$ | | communication threshold |
| $r_t$ | $f_{w^*}(x^*) - f_{w^*}(x_t)$ | instantaneous regret at round $t$ |
| $R_T$ | $\sum_{t=1}^{T} r_t$ | cumulative regret after round $T$ |
| $\Sigma_t$ | equation 6 | covariance matrix at round $t$ |
| $T_0$ | | time horizon in Phase I |
| $T$ | | time horizon in Phase II |
| $\mathcal{U}$ | | uniform distribution |
| $w$ | $w \in \mathcal{W}$ | function parameter |
| $w^*$ | $w^* \in \mathcal{W}$ | true parameter |
| $\hat{w}_0$ | | oracle-estimated parameter after Phase I |
| $\hat{w}_{t,i}$ | equation 5 | updated parameter of client $i$ at round $t$ |
| $\mathcal{W}$ | $\mathcal{W} \subseteq [0,1]^{d_w}$ | parameter space |
| $\mathbf{x}$ | $\mathbf{x} \in \mathcal{X}$ | data point |
| $\mathbf{x}^*$ | | optimal data point |
| $\|\mathbf{x}\|_p$ | $(\sum_{i=1}^{d} |\mathbf{x}_i|^p)^{1/p}$ | $\ell_p$ norm |
| $\|\mathbf{x}\|_A$ | $\sqrt{\mathbf{x}^\top A \mathbf{x}}$ | distance defined by square matrix $A$ |
| $\mathcal{X}$ | $\mathcal{X} \subseteq \mathbb{R}^{d_x}$ | function domain |
| $\mathcal{Y}$ | $\mathcal{Y} = [-F, F]$ | function range |

# B ADDITIONAL DISCUSSIONS

## B.1 MORE DETAILS ON RELATED WORKS

**More details on distributed/federated bandits** For distributed bandits (Korda et al., 2016; Mahadik et al., 2020; Wang et al., 2020; Dubey & Pentland, 2020; Huang et al., 2021), designing an efficient communication strategy is the main focus. Existing algorithms mainly differ in the relations of learning problems solved by the clients (i.e., identical vs., clustered) and the type of communication network (i.e., peer-to-peer (P2P) vs., star-shaped). Korda et al. (2016) studied two problem settings with a P2P communication network: 1) all the clients solve a common linear bandit problem, and 2) the problems are clustered. Mahadik et al. (2020) later proposed an improved algorithm on the second problem setting studied by Korda et al. (2016). However, both works only tried to reduce *per-round* communication, and thus the communication cost is still linear over time. Two follow-up studies considered the setting where all clients solve a common linear bandit problem with time-varying arm set and interact with the environment in a round-robin fashion (Wang et al., 2020; Dubey & Pentland, 2020). Similar to our work, they also used event-triggered communications to obtain a sub-linear communication cost over time. In particular, Wang et al. (2020) considered a star-shaped network and proposed a synchronous communication protocol for all clients to exchange their sufficient statistics via the central server. Dubey & Pentland (2020) extended this synchronous protocol to differentially private LinUCB algorithms under both star-shaped and P2P network. Huang et al. (2021) considered a similar setting but with a fixed arm set and thus proposed a phase-based elimination algorithm. Later, in order to improve robustness against possible stragglers in the system, Li & Wang (2022a); He et al. (2022) proposed asynchronous communication protocols for the federated linear bandit problems. All the works mentioned above consider linear models, where efficient communication is enabled via closed-form model update. The only existing work that studies nonlinear models with iterative optimization is by Li & Wang (2022b), who employed a combination of online and offline regression, with online regression adjusting each client's local model using its newly collected data, and offline (distributed) regression that conducts iterative gradient aggregation over all clients for joint model estimation during global synchronizations.

**Offline federated learning** Another related line of research is federated learning or decentralized machine learning that considers offline supervised learning scenarios (Kairouz et al., 2019). FedAvg (McMahan et al., 2017) has been the most popular algorithm for offline federated learning. However, despite its popularity, several works (Li et al., 2019a; Karimireddy et al., 2020; Mitra et al., 2021) identified that FedAvg suffers from a *client-drift* problem when the clients' data are non-IID (which is an important signature of our case), i.e., local iterates in each client drift towards their local minimum. This leads to a sub-optimal convergence rate of FedAvg, i.e., a sub-linear convergence rate for strongly convex and smooth losses, though a linear convergence rate is expected in this case. To address this, Pathak & Wainwright (2020) proposed an operator splitting procedure to guarantee linear convergence to a neighborhood of the global minimum. Later, Mitra et al. (2021) introduced variance reduction techniques to guarantee exact linear convergence to the global minimum. However, due to the fundamental difference in the learning objectives, they are not suitable for our federated optimization problem: their focus is to collaboratively learn a good *point estimate* over a fixed dataset, i.e., convergence to the minimizer with fewer iterations, while federated bandit learning requires collaborative *confidence set estimation* for efficient regret reduction. This is also reflected by the difference in the design of communication triggering events. For decentralized supervised machine learning, triggering event measuring the change in the learned parameters suffices (Kia et al., 2015; Yi et al., 2018; George & Gurram, 2020), while for federated bandit learning, triggering event needs to measure change in the volume of the confidence set, i.e., uncertainty in the problem space (Wang et al., 2020; Li & Wang, 2022a).

## B.2 SUMMARY OF EXISTING FEDERATED BANDIT ALGORITHMS

Here we provide a summary of existing works in federated bandit optimization in Table 2, which mainly differs in their modeling assumptions, type of communication protocol (synchronous vs asynchronous), as well as theoretical guarantees on regret and communication cost (in terms of $N$ and $T$).

Table 2: Summary of existing federated bandit algorithms.

| Related Works | Modeling Assumption | Protocol | Regret | Communication |
|---|---|---|---|---|
| Wang et al. (2020) | linear | synchronous | $\sqrt{NT}$ | $N^{1.5}$ |
| Li & Wang (2022a) | linear | asynchronous | $\sqrt{NT}$ | $N^2$ |
| He et al. (2022) | linear | asynchronous | $\sqrt{NT} + N$ | $N^2$ |
| Li & Wang (2022b) | generalized linear | synchronous | $\sqrt{NT}$ | $N^2\sqrt{T}$ |
| Li et al. (2022a) | kernel | synchronous | $\sqrt{NT}$ | $N^2$ |
| Li et al. (2022b) | kernel | asynchronous | $\sqrt{NT} + N$ | $N^2$ |
| Dai et al. (2022) | kernel (NTK) | synchronous | $N\sqrt{T}$ | $N^{1.5}T^2$ |
| Ours | nonconvex | synchronous | $\sqrt{NT}$ | $N^{1.5}\sqrt{T}$ |

## C  AUXILIARY LEMMAS

**Lemma 9** (Lemma C.5 of Liu & Wang (2023))**.** *Set $\Sigma_{t,i_t}$ as in equation 6 and suppose Assumptions 1, 2, & 3 hold. Then*

$$\ln\left(\frac{\det \Sigma_{t-1,i_t}}{\det \Sigma_0}\right) \leq d_w \ln\left(1 + \frac{|D_{t,i_t}|C_g^2}{d_w\lambda}\right).$$

**Lemma 10** (Lemma C.6 of Liu & Wang (2023))**.** *Set $\Sigma_{t,i_t}$ as in equation 6 and suppose Assumptions 1, 2, & 3 hold. Then*

$$\sum_{s\in D_{t,i_t}} \nabla f_{x_s}(\hat{w}_0)^\top \Sigma_{s-1,i_t}^{-1} \nabla f_{x_s}(\hat{w}_0) \leq 2\ln(\frac{\Sigma_{t-1,i_t}}{\Sigma_0})$$

**Lemma 11** (Self-normalized bound for vector-valued martingales (Abbasi-yadkori et al., 2011; Agarwal et al., 2021))**.** *Let $\{\eta_i\}_{i=1}^\infty$ be a real-valued stochastic process with corresponding filtration $\{\mathcal{F}_i\}_{i=1}^\infty$ such that $\eta_i$ is $\mathcal{F}_i$ measurable, $\mathbb{E}[\eta_i|\mathcal{F}_{i-1}] = 0$, and $\eta_i$ is conditionally $\sigma$-sub-Gaussian with $\sigma \in \mathbb{R}^+$. Let $\{X_i\}_{i=1}^\infty$ be a stochastic process with $X_i \in \mathcal{H}$ (some Hilbert space) and $X_i$ being $F_t$ measurable. Assume that a linear operator $\Sigma : \mathcal{H} \to \mathcal{H}$ is positive definite, i.e., $x^\top\Sigma x > 0$ for any $x \in \mathcal{H}$. For any $t$, define the linear operator $\Sigma_t = \Sigma_0 + \sum_{i=1}^t X_iX_i^\top$ (here $xx^\top$ denotes outer-product in $\mathcal{H}$). With probability at least $1 - \delta$, we have for all $t \geq 1$:*

$$\left\|\sum_{i=1}^t X_i\eta_i\right\|_{\Sigma_t^{-1}}^2 \leq \sigma^2 \ln\left(\frac{\det(\Sigma_t)\det(\Sigma_0)^{-1}}{\delta^2}\right). \tag{7}$$

**Lemma 12** (Risk Bounds for $\epsilon$-approximation of ERM Estimator)**.** *Given a dataset $\{x_s, y_s\}_{s=1}^t$ where $y_s$ is generated from equation 1, and $f^\star$ is the underlying true function. Let $\hat{f}_t$ be an ERM estimator taking values in $\mathcal{F}$ where $\mathcal{F}$ is a finite set and $\mathcal{F} \subset \{f : [0,1]^d \to [-F, F]\}$ for some $F \geq 1$. $\tilde{f}_t \in \mathcal{F}$ denotes its $\epsilon$-approximation. Then with probability at least $1 - \delta$, $\tilde{f}$ satisfies that*

$$\mathbb{E}\big[(\tilde{f}_t - f_0)^2\big] \leq \frac{1+\alpha}{1-\alpha}\big(\inf_{f\in\mathcal{F}} \mathbb{E}[(f - f_0)^2] + \frac{F^2\ln(|\mathcal{F}|)\ln(2)}{t\alpha}\big) + \frac{2\ln(2/\delta)}{t\alpha} + \frac{\epsilon}{1-\alpha}$$

*for all $\alpha \in (0, 1]$ and $\epsilon \geq 0$.*

*Proof of Lemma 12.* Define the risk and empirical risk function as

$$R(f) := \mathbb{E}_{X,Y}[(f(X) - Y)^2],$$

$$\hat{R}(f) := \frac{1}{t}\sum_{s=1}^t (f(x_s) - y_s)^2.$$

By definition,

$$f^\star = \mathbb{E}[Y|X = x] = \arg\min_{f\in\mathcal{F}} R(f).$$

Denote the excess risk and the empirical excess risk as

$$\mathcal{E}(f) := R(f) - R(f^\star),$$
$$\hat{\mathcal{E}}(f) := \hat{R}(f) - \hat{R}(f^\star).$$

Following the same steps in Nowak [2007], we have

$$(1-\alpha)\mathcal{E}(f) \le \hat{\mathcal{E}}(f) + \frac{c(f)\ln 2 + \ln 1/\delta}{\alpha t} = \hat{R}(f) - \hat{R}(f^\star) + \frac{c(f)\ln 2 + \ln 1/\delta}{\alpha t},$$

where penalties $c(f)$ are positive numbers assigned to each $f \in \mathcal{F}$ that satisfies $\sum_{f\in\mathcal{F}} 2^{-c(f)} \le 1$. We set it to $c(f) = F^2 \ln(|\mathcal{F}|)$ according to Lemma 4 of Schmidt-Hieber (2020). Now recall that, we have denoted $\hat{f}_t$ as the ERM estimator, i.e., $\hat{f}_t := \arg\min_{f\in\mathcal{F}} \hat{R}(f)$, and $\tilde{f}_t$ as its $\epsilon$-approximation, such that

$$\hat{R}(\tilde{f}_t) - \hat{R}(\hat{f}_t) \le \epsilon.$$

Therefore, we have

$$(1-\alpha)\mathcal{E}(\tilde{f}_t) \le \hat{\mathcal{E}}(\tilde{f}_n) + \frac{F^2\ln(|\mathcal{F}|)\ln 2 + \ln 1/\delta}{\alpha t}$$
$$\le \hat{\mathcal{E}}(\hat{f}_t) + \frac{F^2\ln(|\mathcal{F}|)\ln 2 + \ln 1/\delta}{\alpha t} + \epsilon$$
$$\le \hat{\mathcal{E}}(f_t^\star) + \frac{F^2\ln(|\mathcal{F}|)\ln 2 + \ln 1/\delta}{\alpha t} + \epsilon$$

Due to Craig-Bernstein inequality, we have

$$\hat{\mathcal{E}}(f_t^\star) \le \mathcal{E}(f_t^\star) + \alpha\mathcal{E}(f_t^\star) + \frac{\ln 1/\delta}{\alpha t}.$$

Combining the two inequalities above, we have

$$\mathcal{E}(\tilde{f}_t) \le \frac{1+\alpha}{1-\alpha}\mathcal{E}(f_t^\star) + \frac{1}{1-\alpha}\frac{F^2\ln(|\mathcal{F}|)\ln 2 + 2\ln 1/\delta}{\alpha t} + \frac{1}{1-\alpha}\epsilon.$$

$\square$

## D   COMPLETE PROOFS

### D.1   PROOF OF LEMMA 6

*Proof of Lemma 6.* The regression oracle lemma establishes on Lemma 12 which works only for finite function class. In order to work with our continuous parameter class W, we need $\varepsilon$-covering number argument. First, let $\tilde{w}, \tilde{\mathcal{W}}$ denote the $\epsilon$-approximation of ERM parameter and finite parameter class after applying covering number argument on $\mathcal{W}$. By Lemma 12, we find that with probability at least $1 - \delta/2$,

$$\mathbb{E}_{x\sim\mathcal{U}}[(f_x(\tilde{w}) - f_x(w^\star))^2] \le \frac{1+\alpha}{1-\alpha}\Big(\inf_{w\in\tilde{\mathcal{W}}\cup\{w^\star\}} \mathbb{E}_{x\sim\mathcal{U}}[(f_x(w) - f_x(w^\star))^2] + \frac{F^2\ln(|\mathcal{W}|)\ln(2)}{T_0\alpha}\Big)$$
$$+ \frac{\epsilon}{1-\alpha} + \frac{2\ln(4/\delta)}{T_0\alpha}$$
$$\le \frac{1+\alpha}{1-\alpha}\Big(\frac{F^2\ln(|\tilde{\mathcal{W}}|)\ln(2)}{T_0\alpha}\Big) + \frac{\epsilon}{1-\alpha} + \frac{2\ln(4/\delta)}{T_0\alpha}$$

where the second inequality is due to Assumption 1. Our parameter class $\mathcal{W} \subseteq [0,1]^{d_w}$, so $\ln(|\tilde{\mathcal{W}}|) = \ln(1/\epsilon^{d_w}) = d_w\ln(1/\varepsilon)$ and the new upper bound is that with probability at least $1 - \delta/2$,

$$\mathbb{E}_{x\sim\mathcal{U}}[(f_x(\tilde{w}) - f_x(w^\star))^2] \le C''\Big(\frac{d_w F^2\ln(1/\varepsilon)}{T_0} + \frac{\ln(2/\delta)}{T_0} + \epsilon\Big)$$

where $C''$ is a universal constant obtained by choosing $\alpha = 1/2$. Note $\tilde{w}$ is the parameter in $\tilde{W}$ after discretization, not our target parameter $\tilde{w}_0 \in \mathcal{W}$. By $(a+b)^2 \le 2a^2 + 2b^2$,

$$\mathbb{E}_{x \sim \mathcal{U}}[(f_x(\hat{w}_0) - f_x(w^\star))^2] \le 2\mathbb{E}_{x \sim \mathcal{U}}[(f_x(\hat{w}_0) - f_x(\tilde{w}))^2] + 2\mathbb{E}_{x \sim \mathcal{U}}[(f_x(\tilde{w}) - f_x(w^\star))^2]$$

$$\le 2\varepsilon^2 C_h^2 + 2C''\Big(\frac{d_w F^2 \ln(1/\varepsilon)}{T_0} + \frac{\ln(2/\delta)}{T_0} + \epsilon\Big)$$

where the second line applies Lemma 12, discretization error $\varepsilon$, and Assumption 2. By choosing $\varepsilon = 1/(2\sqrt{T_0 C_h^2})$, and $\epsilon = 1/(2C''T_0)$ we get

$$(18) = \frac{2}{T_0} + \frac{C'' d_w F^2 \ln(T_0 C_h^2)}{T_0} + \frac{2C'' \ln(2/\delta)}{T_0} \le C' \frac{d_w F^2 \ln(T_0 C_h^2) + \ln(2/\delta)}{T_0}$$

where we can take $C' = 2C''$ (assuming $2 < C'' d_w F^2 \log(T_0 C_h^2)$). The proof completes by defining $\iota$ as the logarithmic term depending on $T_0, C_h, 2/\delta$. $\qquad\square$

## D.2 CONFIDENCE ANALYSIS

**Lemma 13** (Restatement of Lemma 8). *Set $\hat{w}_{t,i_t}, \Sigma_{t,i_t}$ as in eq. equation 6, equation 5. Set $\beta_{t,i_t}$ as*

$$\beta_{t,i_t} = \tilde{\Theta}\left(d_w \sigma^2 + \frac{d_w F^2}{\mu} + \frac{d_w^3 F^4}{\mu^2}\right).$$

*Suppose Assumptions 1, 2, & 3 hold and choose $T_0 = \sqrt{NT}, \lambda = C_\lambda \sqrt{NT}$. Then $\forall t \in [NT]$ in Phase II of Algorithm 1, $\|\hat{w}_{t,i_t} - w^*\|_{\Sigma_{t,i_t}}^2 \le \beta_{t,i_t}$, with probability at least $1 - \delta$.*

*Proof.* The proof has two steps. First we obtain the closed form solution of $\hat{w}_{t,i_t}$. Next we prove the upper bound of $\|\hat{w}_{t,i_t} - w^*\|_{\Sigma_{t,i_t}}^2$ matches our choice of $\beta_{t,i_t}$.

**Step 1: Closed form solution of $\hat{w}_{t,i_t}$.** Let $\nabla$ denote $\nabla f_{\mathbf{x}_s}(\hat{w}_0)$ in this proof.

Recall $\hat{w}_{t,i_t}$ is estimated by solving the following optimization problem:

$$\hat{w}_{t,i_t} = \Sigma_{t,i_t}^{-1} b_{t,i_t} + \lambda \Sigma_{t,i_t}^{-1} \hat{w}_0$$

$$= \arg\min_w \frac{\lambda}{2}\|w - \hat{w}_0\|_2^2 + \frac{1}{2}\sum_{s \in \mathcal{D}_t(i_t)} ((w - \hat{w}_0)^\top \nabla + f_{\mathbf{x}_s}(\hat{w}_0) - y_s)^2$$

The optimal criterion for the objective function is

$$0 = \lambda(\hat{w}_{t,i_t} - \hat{w}_0) + \sum_{s \in \mathcal{D}_t(i_t)} ((\hat{w}_{t,i_t} - \hat{w}_0)^\top \nabla + f_{\mathbf{x}_s}(\hat{w}_0) - y_s)\nabla.$$

Rearrange the equation and we have

$$\lambda(\hat{w}_{t,i_t} - \hat{w}_0) + \sum_{s \in \mathcal{D}_t(i_t)} (\hat{w}_{t,i_t} - \hat{w}_0)^\top \nabla\nabla = \sum_{s \in \mathcal{D}_t(i_t)} (y_s - f_{\mathbf{x}_s}(\hat{w}_0))\nabla,$$

$$\lambda(\hat{w}_{t,i_t} - \hat{w}_0) + \sum_{s \in \mathcal{D}_t(i_t)} (\hat{w}_{t,i_t} - \hat{w}_0)^\top \nabla\nabla = \sum_{s \in \mathcal{D}_t(i_t)} (y_s - f_{\mathbf{x}_s}(w^*) + f_{\mathbf{x}_s}(w^*) - f_{\mathbf{x}_s}(\hat{w}_0))\nabla,$$

$$\lambda(\hat{w}_{t,i_t} - \hat{w}_0) + \sum_{s \in \mathcal{D}_t(i_t)} \hat{w}_{t,i_t}^\top \nabla\nabla = \sum_{s \in \mathcal{D}_t(i_t)} (\hat{w}_0^\top \nabla + \eta_s + f_{\mathbf{x}_s}(w^*) - f_{\mathbf{x}_s}(\hat{w}_0))\nabla,$$

$$\hat{w}_{t,i_t}\left(\lambda \mathbf{I} + \sum_{s \in \mathcal{D}_t(i_t)} \nabla\nabla^\top\right) - \lambda\hat{w}_0 = \sum_{s \in \mathcal{D}_t(i_t)} (\hat{w}_0^\top \nabla + \eta_s + f_{\mathbf{x}_s}(w^*) - f_{\mathbf{x}_s}(\hat{w}_0))\nabla,$$

$$\hat{w}_{t,i_t}\Sigma_{t,i_t} = \lambda\hat{w}_0 + \sum_{s \in \mathcal{D}_t(i_t)} (\hat{w}_0^\top \nabla + \eta_s + f_{\mathbf{x}_s}(w^*) - f_{\mathbf{x}_s}(\hat{w}_0))\nabla,$$

where the second line is by removing and adding back $f_{x_s}(w^*)$, the third line is due to definition of observation noise $\eta$ and the last line is by our choice of $\Sigma_{t,i_t}$ (eq. equation 6). Now we have the closed form solution of $\hat{w}_{t,i_t}$:

$$\hat{w}_{t,i_t} = \lambda \Sigma_{t,i_t}^{-1} \hat{w}_0 + \Sigma_{t,i_t}^{-1} \sum_{s \in \mathcal{D}_t(i_t)} (\hat{w}_0^\top \nabla + \eta_s + f_{\mathbf{x}_s}(w^*) - f_{\mathbf{x}_s}(\hat{w}_0))\nabla,$$

where $\nabla$ denotes $\nabla f_{\mathbf{x}_s}(\hat{w}_0)$. Then $\hat{w}_{t,i_t} - w^*$ can be written as

$$\hat{w}_{t,i_t} - w^* = \Sigma_{t,i_t}^{-1} \left( \sum_{s \in \mathcal{D}_t(i_t)} \nabla(\nabla^\top \hat{w}_0 + \eta_s + f_{\mathbf{x}_s}(w^*) - f_{\mathbf{x}_s}(\hat{w}_0)) \right) + \lambda \Sigma_{t,i_t}^{-1} \hat{w}_0 - \Sigma_{t,i_t}^{-1} \Sigma_{t,i_t} w^*$$

$$= \Sigma_{t,i_t}^{-1} \left( \sum_{s \in \mathcal{D}_t(i_t)} \nabla(\nabla^\top \hat{w}_0 + \eta_s + f_{\mathbf{x}_s}(w^*) - f_{\mathbf{x}_s}(\hat{w}_0)) \right) + \lambda \Sigma_{t,i_t}^{-1}(\hat{w}_0 - w^*)$$

$$- \Sigma_{t,i_t}^{-1} \left( \sum_{s \in \mathcal{D}_t(i_t)} \nabla \nabla^\top \right) w^*$$

$$= \Sigma_{t,i_t}^{-1} \left( \sum_{s \in \mathcal{D}_t(i_t)} \nabla(\nabla^\top (\hat{w}_0 - w^*) + \eta_s + f_{\mathbf{x}_s}(w^*) - f_{\mathbf{x}_s}(\hat{w}_0)) \right) + \lambda \Sigma_{t,i_t}^{-1}(\hat{w}_0 - w^*)$$

$$= \Sigma_{t,i_t}^{-1} \left( \sum_{s \in \mathcal{D}_t(i_t)} \nabla \frac{1}{2} \|\hat{w}_0 - w^*\|_{\nabla^2 f_{\mathbf{x}_s}(\tilde{w})}^2 \right) + \Sigma_{t,i_t}^{-1} \left( \sum_{s \in \mathcal{D}_t(i_t)} \nabla \eta_s \right) + \lambda \Sigma_{t,i_t}^{-1}(\hat{w}_0 - w^*),$$

$$(8)$$

where the second line is again by our choice of $\Sigma_t$ and the last equation is by the second order Taylor's theorem of $f_{\mathbf{x}_s}(w^*)$ at $\hat{w}_0$ where $\tilde{w}$ lies between $w^*$ and $\hat{w}_0$.

**Step 2: Upper bound of $\|\hat{w}_{t,i_t} - w^*\|_{\Sigma_{t,i_t}}^2$.** Multiply both sides of eq. equation 8 by $\Sigma_{t,i_t}^{\frac{1}{2}}$ and we have

$$\Sigma_{t,i_t}^{\frac{1}{2}}(\hat{w}_{t,i_t} - w^*) \leq \frac{1}{2} \Sigma_{t,i_t}^{-\frac{1}{2}} \left( \sum_{s \in \mathcal{D}_t(i_t)} \nabla f_{\mathbf{x}_s}(\hat{w}_0) \|\hat{w}_0 - w^*\|_{\nabla^2 f_{\mathbf{x}_s}(\tilde{w})}^2 \right)$$

$$+ \Sigma_{t,i_t}^{-\frac{1}{2}} \left( \sum_{s \in \mathcal{D}_t(i_t)} \nabla f_{\mathbf{x}_s}(\hat{w}_0) \eta_s \right) + \lambda \Sigma_{t,i_t}^{-\frac{1}{2}}(\hat{w}_0 - w^*).$$

Take square of both sides and by inequality $(a + b + c)^2 \leq 4a^2 + 4b^2 + 4c^2$ we obtain

$$\|\hat{w}_{t,i_t} - w^*\|_{\Sigma_{t,i_t}}^2 \leq 4 \left\| \sum_{s \in \mathcal{D}_t(i_t)} \nabla f_{\mathbf{x}_s}(\hat{w}_0) \eta_s \right\|_{\Sigma_{t,i_t}^{-1}}^2 + 4\lambda^2 \|\hat{w}_0 - w^*\|_{\Sigma_{t,i_t}^{-1}}^2$$

$$+ \left\| \sum_{s \in \mathcal{D}_t(i_t)} \nabla f_{\mathbf{x}_s}(\hat{w}_0) \|\hat{w}_0 - w^*\|_{\nabla^2 f_{\mathbf{x}_s}(\tilde{w})}^2 \right\|_{\Sigma_{t,i_t}^{-1}}^2. \qquad (9)$$

The remaining job is to bound three terms in eq. equation 9 individually. The first term of eq. equation 9 can be bounded as

$$4 \left\| \sum_{s \in \mathcal{D}_t(i_t)} \nabla f_{\mathbf{x}_s}(\hat{w}_0) \eta_s \right\|_{\Sigma_{t,i_t}^{-1}}^2 \leq 4\sigma^2 \log \left( \frac{\det(\Sigma_{t,i_t}) \det(\Sigma_0)^{-1}}{\delta_t^2} \right)$$

$$\leq 4\sigma^2 \left( d_w \log \left( 1 + \frac{i C_g^2}{d_w \lambda} \right) + \log \left( \frac{\pi^2 t^2}{3\delta} \right) \right)$$

$$\leq 4 d_w \sigma^2 \iota',$$

where the second inequality is due to self-normalized bound for vector-valued martingales (Lemma 11 in Appendix C) and Lemma 7, the second inequality is by Lemma 9 and our choice of $\delta_i = 3\delta/(\pi^2 i^2)$, and the last inequality is by defining $\iota'$ as the logarithmic term depending on $i, d_w, C_g, 1/\lambda, 2/\delta$ (with probability $> 1 - \delta/2$). The choice of $\delta_i$ guarantees the total failure probability over $t$ rounds is no larger than $\delta/2$.

Using Lemma 7, the second term in eq. equation 9 is bounded as

$$4\lambda^2 \|\hat{w}_0 - w^*\|^2_{\Sigma^{-1}_{t,i_t}} \le \frac{4\lambda C d_w F^2 \iota}{\mu T_0}.$$

Again using Lemma 7 and Assumption 3, the third term of eq. equation 9 can be bounded as

$$\left\| \sum_{s \in \mathcal{D}_t(i_t)} \nabla f_{\mathbf{x}_s}(\hat{w}_0) \|\hat{w}_0 - w^*\|^2_{\nabla^2 f_{\mathbf{x}_s}(\tilde{w})} \right\|^2_{\Sigma^{-1}_{t,i_t}}$$

$$\le \left\| \frac{CC_h d_w F^2 \iota}{\mu T_0} \sum_{s \in \mathcal{D}_t(i_t)} \nabla f_{\mathbf{x}_s}(\hat{w}_0) \right\|^2_{\Sigma^{-1}_{t,i_t}}$$

$$= \frac{C^2 C_h^2 d_w^2 F^4 \iota^2}{\mu^2 T_0^2} \left\| \sum_{s \in \mathcal{D}_t(i_t)} \nabla f_{\mathbf{x}_s}(\hat{w}_0) \right\|^2_{\Sigma^{-1}_{t,i_t}}$$

$$= \frac{C^2 C_h^2 d_w^2 F^4 \iota^2}{\mu^2 T_0^2} \left( \sum_{s \in \mathcal{D}_t(i_t)} \nabla f_{\mathbf{x}_s}(\hat{w}_0) \right)^\top \Sigma^{-1}_{t,i_t} \left( \sum_{s' \in \mathcal{D}_t(i_t)} \nabla f_{\mathbf{x}_{s'}}(\hat{w}_0) \right).$$

Rearrange the summation and we can write

$$\frac{C^2 C_h^2 d_w^2 F^4 \iota^2}{\mu^2 T_0^2} \left( \sum_{s \in \mathcal{D}_t(i_t)} \nabla f_{\mathbf{x}_s}(\hat{w}_0) \right)^\top \Sigma^{-1}_{t,i_t} \left( \sum_{s' \in \mathcal{D}_t(i_t)} \nabla f_{\mathbf{x}_{s'}}(\hat{w}_0) \right)$$

$$= \frac{C^2 C_h^2 d_w^2 F^4 \iota^2}{\mu^2 T_0^2} \sum_{s \in \mathcal{D}_t(i_t)} \sum_{s' \in \mathcal{D}_t(i_t)} \nabla f_{\mathbf{x}_s}(\hat{w}_0)^\top \Sigma^{-1}_{t,i_t} \nabla f_{\mathbf{x}_{s'}}(\hat{w}_0)$$

$$\le \frac{C^2 C_h^2 d_w^2 F^4 \iota^2}{\mu^2 T_0^2} \sum_{s \in \mathcal{D}_t(i_t)} \sum_{s' \in \mathcal{D}_t(i_t)} \|\nabla f_{\mathbf{x}_s}(\hat{w}_0)\|_{\Sigma^{-1}_{t,i_t}} \|\nabla f_{\mathbf{x}_{s'}}(\hat{w}_0)\|_{\Sigma^{-1}_{t,i_t}}$$

$$= \frac{C^2 C_h^2 d_w^2 F^4 \iota^2}{\mu^2 T_0^2} \left( \sum_{s \in \mathcal{D}_t(i_t)} \|\nabla f_{\mathbf{x}_s}(\hat{w}_0)\|_{\Sigma^{-1}_{t,i_t}} \right) \left( \sum_{s' \in \mathcal{D}_t(i_t)} \|\nabla f_{\mathbf{x}_{s'}}(\hat{w}_0)\|_{\Sigma^{-1}_{t,i_t}} \right)$$

$$= \frac{C^2 C_h^2 d_w^2 F^4 \iota^2}{\mu^2 T_0^2} \left( \sum_{s \in \mathcal{D}_t(i_t)} \|\nabla f_{\mathbf{x}_s}(\hat{w}_0)\|_{\Sigma^{-1}_{t,i_t}} \right)^2$$

$$\le \frac{C^2 C_h^2 d_w^2 F^4 \iota^2}{\mu^2 T_0^2} \left( \sum_{s \in \mathcal{D}_t(i_t)} 1 \right) \left( \sum_{s \in \mathcal{D}_t(i_t)} \|\nabla f_{\mathbf{x}_s}(\hat{w}_0)\|^2_{\Sigma^{-1}_{t,i_t}} \right)$$

$$\le \frac{C^2 C_h^2 d_w^3 F^4 t \iota'' \iota^2}{\mu^2 T_0^2},$$

where the second last inequality is due to Cauchy-Schwarz inequality and the last inequality is by Lemma 10.

Finally, put three bounds together and we have

$$\|\hat{w}_{t,i_t} - w^*\|^2_{\Sigma^{-1}_{t,i_t}} \leq 4d_w\sigma^2\iota' + \frac{4\lambda C d_w F^2\iota}{\mu T_0} + \frac{C^2 C_h^2 d_w^3 F^4 t\iota''\iota^2}{\mu^2 T_0^2}$$
$$\leq O\left(d_w\sigma^2\iota' + \frac{d_w F^2\iota}{\mu} + \frac{d_w^3 F^4 t\iota''\iota^2}{\mu^2 NT}\right),$$

where the last inequality is by our choices of $\lambda = C_\lambda\sqrt{NT}, T_0 = \sqrt{NT}$. Therefore, our choice of

$$\beta_{t,i_t} = \tilde{\Theta}\left(d_w\sigma^2 + \frac{d_w F^2}{\mu} + \frac{d_w^3 F^4}{\mu^2}\right)$$

guarantees that $w^*$ is always contained in $\text{Ball}_t$ with probability $1 - \delta$. □

### D.3    CUMULATIVE REGRET AND COMMUNICATION COST IN PHASE II

**Cumulative Regret in Phase II**    Thanks to the confidence set established in Lemma 8, Phase II of Algorithm 1 can operate in a similar way as existing works in federated linear bandits (Wang et al., 2020; Dubey & Pentland, 2020), while allowing for a much wider choices of models. The main difference is that, the regret of their work depends on the matrix constructed using context vectors $\mathbf{x}_s$ for $s = 1, 2, \dots$ for the selected points, while ours rely on the matrix constructed using gradients' w.r.t. the shared model $\hat{w}_0$. In the following paragraphs, we first establish the relation between instantaneous regret $r_t$ and matrix $\Sigma_{t-1,i_t}$, and then analyze the regret of Algorithm 1.

Though, compared with Liu & Wang (2023), we are using a different way of constructing the confidence ellipsoid, which is given in Lemma 8. Their Lemma 5.4, which is given below, still holds, because it only requires $\text{Ball}_{t-1,i_t}$ to be a valid confidence set, and that Assumption 2 holds.

**Lemma 14** (Instantaneous regret bound [Lemma 5.4 of Liu & Wang (2023)). *Under the same condition as Lemma 8,in Phase II of Algorithm 1, for all $t \in [NT]$, we have*

$$r_t \leq 2\sqrt{\beta_{t-1,i_t}}\|\nabla f_{\mathbf{x}_t}(\hat{w}_0)\|_{\Sigma^{-1}_{t-1,i_t}} + 2\beta_{t,i_t}C_h/\lambda,$$

*with probability at least $1 - \delta$.*

Denote the total number of global synchronizations (total number of times the event in line 6 of Algorithm 1 is true) over time horizon $T$ as $P \in [0, NP]$. Then we use $t_p$ for $p \in [P]$ to denote the time step when the $p$-th synchronization happens (define $t_0 = 0$), and refer to the sequence of time steps in-between two consecutive synchronizations as an epoch, i.e., the $p$-th epoch is $[t_{p-1} + 1, t_p]$. Similar to Wang et al. (2020); Dubey & Pentland (2020); Li & Wang (2022b), for the cumulative regret analysis in Phase II, we decompose $P$ epochs into good and bad epochs, and then analyze them separately.

Specifically, consider an imaginary centralized agent that has immediate access to each data point in the learning system, and we let this centralized agent executes the same model update rule and arm selection rule as in line 3-5 of Algorithm 1. Then the covariance matrix maintained by this agent can be defined as $\Sigma_t = \sum_{s=1}^t \nabla f_{x_s}(\hat{w}_0)\nabla f_{x_s}(\hat{w}_0)^\top$ for $t \in [NT]$. The $p$-th epoch is called a good epoch if

$$\ln\left(\frac{\det(\Sigma_{t_p})}{\det(\Sigma_{t_{p-1}})}\right) \leq 1,$$

otherwise it is a bad epoch. Note that based on Lemma 9, we have $\ln(\det(\Sigma_{NT})/\det(\lambda I)) \leq d_w \log\left(1 + \frac{NTC_g^2}{d_w\lambda}\right) := R$. Since $\ln\left(\frac{\det(\Sigma_{t_1})}{\det(\lambda I)}\right) + \ln\left(\frac{\det(\Sigma_{t_2})}{\det(\Sigma_{t_1})}\right) + \cdots + \ln\left(\frac{\det(\Sigma_{NT})}{\det(\Sigma_{t_B})}\right) = \ln\left(\frac{\det(\Sigma_{NT})}{\det(\lambda I)}\right) \leq R$, and due to the pigeonhole principle, there can be at most $R$ bad epochs.

Now consider some good epoch $p$. By definition, we have $\frac{\det(\Sigma_{t-1})}{\det(\Sigma_{t-1,i_t})} \leq \frac{\det(\Sigma_{t_p})}{\det(\Sigma_{t_{p-1}})} \leq e$, for any $t \in [t_{p-1}+1, t_p]$. Therefore, if the instantaneous regret $r_t$ is incurred during a good epoch, we have

$$
\begin{aligned}
r_t &\leq 2\sqrt{\beta_{t-1,i_t}}\|\nabla f_{x_t}(\hat{w}_0)\|_{\Sigma_{t-1,i_t}^{-1}} + \frac{2\beta_{t-1,i_t}C_h}{\lambda} \\
&\leq 2\sqrt{\beta_{t-1,i_t}}\|\nabla f_{x_t}(\hat{w}_0)\|_{\Sigma_{t-1}^{-1}}\sqrt{\|\nabla f_{x_t}(\hat{w}_0)\|_{\Sigma_{t-1,i_t}^{-1}}/\|\nabla f_{x_t}(\hat{w}_0)\|_{\Sigma_{t-1}^{-1}}} + \frac{2\beta_{t-1,i_t}C_h}{\lambda} \\
&= 2\sqrt{\beta_{t-1,i_t}}\|\nabla f_{x_t}(\hat{w}_0)\|_{\Sigma_{t-1}^{-1}}\sqrt{\frac{\det(\Sigma_{t-1})}{\det(\Sigma_{t-1,i_t})}} + \frac{2\beta_{t-1,i_t}C_h}{\lambda} \\
&\leq 2\sqrt{e}\sqrt{\beta_{t-1,i_t}}\|\nabla f_{x_t}(\hat{w}_0)\|_{\Sigma_{t-1}^{-1}} + \frac{2\beta_{t-1,i_t}C_h}{\lambda}
\end{aligned}
$$

where the first inequality is due to Lemma 14, and the last inequality is due to the definition of good epoch, i.e., $\frac{\det(\Sigma_{t-1})}{\det(\Sigma_{t-1,i_t})} \leq \frac{\det(\Sigma_{t_p})}{\det(\Sigma_{t_{p-1}})} \leq e$.

This suggests the instantaneous regret $r_t$ incurred in a good epoch is at most $\sqrt{e}$ times of that incurred by the imaginary centralized agent that runs GO-UCB algorithm of Liu & Wang (2023). Therefore, the cumulative regret incurred in good epochs of Phase II, denoted as $R_{\text{good}}$ is

$$
\begin{aligned}
R_{\text{good}} &= \sum_{p=1}^{P}\mathbb{1}\{\ln\left(\frac{\det(\Sigma_{t_p})}{\det(\Sigma_{t_{p-1}})}\right) \leq 1\}\sum_{t=t_{p-1}}^{t_p} r_t \leq \sum_{t=1}^{NT} r_t \leq \sqrt{NT\sum_{t=1}^{NT} r_t^2} \\
&\leq \sqrt{NT}\sqrt{16e\beta_{NT}d_w\ln(1+\frac{NTC_g^2}{d_w\lambda}) + \frac{8\beta_{NT}^2C_h^2NT}{\lambda^2}}
\end{aligned}
$$

where the last inequality is due to $(a+b)^2 \leq 2a^2+2b^2$, Lemma 9, and Lemma 10. Note that $\beta_{NT} = O(\frac{d_w^3 F^4}{\mu^2})$ according to Lemma 8. By setting $\lambda = C_\lambda\sqrt{NT}$, we have $R_{\text{good}} = \tilde{O}(\frac{d_w^3 F^4\sqrt{NT}}{\mu^2})$.

Consider some bad epoch $p$, we can upper bound the cumulative regret incurred by all $N$ clients in this epoch $p$ as

$$
\begin{aligned}
\sum_{t=t_{p-1}+1}^{t_p} r_t &= \sum_{i=1}^{N}\sum_{t \in D_{t_p,i}\backslash D_{t_{p-1},i}} r_t \leq \sum_{i=1}^{N}\sum_{t \in D_{t_p,i}\backslash D_{t_{p-1},i}}\left(2\sqrt{\beta_{t-1,i_t}}\|\nabla f_{x_t}(\hat{w}_0)\|_{\Sigma_{t-1,i_t}^{-1}} + \frac{2\beta_{t-1,i_t}C_h}{\lambda}\right) \\
&\leq \sum_{i=1}^{N}\sqrt{(|D_{t_p,i}|-|D_{t_{p-1},i}|)8\beta_{NT}\sum_{t \in D_{t_p,i}\backslash D_{t_{p-1},i}}\|\nabla f_{x_t}(\hat{w}_0)\|_{\Sigma_{t-1,i_t}^{-1}}^2 + 8\beta_{NT}^2C_h^2/C_\lambda^2} \\
&\leq \sum_{i=1}^{N}\sqrt{16\beta_{NT}\gamma + 8\beta_{NT}^2C_h^2/C_\lambda^2}
\end{aligned}
$$

where the first inequality is due to Lemma 14, the second is due to Cauchy-Schwartz inequality and $(a+b)^2 \leq 2a^2+2b^2$, and the last is due to Lemma 10 and event-trigger with threshold $\gamma$ in line 6 of Algorithm 1.

Since there can be at most $R = d_w\log\left(1+\frac{NTC_g^2}{d_w\lambda}\right)$ bad epochs, the cumulative regret incurred in bad epochs of Phase II, denoted as $R_{\text{bad}}$ is $\tilde{O}(Nd_w\sqrt{\frac{d_w^3 F^4}{\mu^2}}\sqrt{\gamma} + Nd_w\frac{d_w^3 F^4}{\mu^2})$. By setting communication threshold $\gamma = \frac{d_w F^4 T}{\mu^2 N}$, we have $R_{\text{bad}} = \tilde{O}(\frac{d_w^3 F^4\sqrt{NT}}{\mu^2} + \frac{d_w^4 F^4 N}{\mu^2})$. Combining cumulative regret incurred during both good and bad epochs, we have

$$
R_{\text{phase II}} = R_{\text{good}} + R_{\text{bad}} = \tilde{O}\left(\frac{d_w^3 F^4}{\mu^2}\sqrt{NT} + \frac{d_w^4 F^4}{\mu^2}N\right).
$$

**Communication Cost in Phase II** Consider some $\alpha > 0$. By pigeon-hole principle, there can be at most $\lceil\frac{NT}{\alpha}\rceil$ epochs with length (total number of time steps) longer than $\alpha$. Then consider some epoch with less than $\alpha$ time steps. We denote the first time step of this epoch as $t_s$ and

the last as $t_e$, i.e., $t_e - t_s < \alpha$. Since the users appear in a round-robin manner, the number of interactions for any user $i \in [N]$ satisfies $|D_{t_e,i}| - |D_{t_s,i}| < \frac{\alpha}{N}$. Due to the event-triggered in line 6 of Algorithm 1, we have $\log \frac{\det(\Sigma_{t_e})}{\det(\Sigma_{t_s})} > \frac{\gamma N}{\alpha}$. Using the pigeonhole principle again, we know that the number of epochs with less than $\alpha$ time steps is at most $\lceil \frac{R\alpha}{\gamma N} \rceil$. Therefore, the total number of synchronizations $P \leq \lceil \frac{NT}{\alpha} \rceil + \lceil \frac{R\alpha}{\gamma N} \rceil$, and the RHS can be minimized by choosing $\alpha = N\sqrt{\gamma T/R}$, so that $P \leq 2\sqrt{TR/\gamma}$. With $\gamma = \frac{d_w F^4 T}{\mu^2 N}$, $P \leq 2\sqrt{\frac{N\log(1+NTC_g^2/(d_w\lambda))\mu^2}{F^4}} = \tilde{O}(\sqrt{N\mu^2/F^4})$. At each global synchronization, Algorithm 1 incurs $2N(d_w^2 + d_w)$ communication cost to update the statistics. Therefore, $C_{\text{phase II}} = P \cdot 2N(d_w^2 + d_w) = (N^{1.5}d_w^2\mu/F^2)$.

# E  EXPERIMENT SETUP & ADDITIONAL RESULTS

**Synthetic dataset experiment setup**  Here we provide more details about the experiment setup on synthetic dataset in Section 6. Specifically, we compared all the algorithms on the following two synthetic functions

$$f_1(\mathbf{x}) = -\sum_{i=1}^{4} \bar{\alpha}_i \exp(-\sum_{j=1}^{6} \bar{A}_{ij}(\mathbf{x}_j - \bar{P}_{ij})^2),$$

$$f_2(\mathbf{x}) = 0.1\sum_{i=1}^{8} \cos(5\pi\mathbf{x}_i) - \sum_{i=1}^{8} \mathbf{x}_i^2.$$

Both are popular synthetic functions for Bayesian optimization benchmarking[1]. The 6-dimensional function $f_1$ is called Hartmann function, where

$$\bar{\alpha} = [1.0, 1.2, 3.0, 3.2], \bar{A} = \begin{bmatrix} 10, 3, 17, 3.5, 1.7, 8 \\ 0.05, 10, 17, 0.1, 8, 14 \\ 3, 3.5, 1.7, 10, 17, 8 \\ 17, 8, 0.05, 10, 0.1, 14 \end{bmatrix}, \bar{P} = \begin{bmatrix} 1312, 1696, 5569, 124, 8283, 5886 \\ 2329, 4135, 8307, 3736, 1004, 9991 \\ 2348, 1451, 3522, 2883, 3047, 6650 \\ 4047, 8828, 8732, 5743, 1091, 381 \end{bmatrix}.$$

And the 8-dimensional function $f_2$ is a cosine mixture test function, which is named Cosine8. To be compatible with the discrete candidate set setting assumed in prior works (Wang et al., 2020; Li & Wang, 2022b; Li et al., 2022a), we generate the decision set $\mathcal{X}$ for the optimization of $f_1$ by uniformly sampling 50 data points from $[0,1]^6$, and similarly for the optimization of $f_2$, 50 data points from $[-1,1]^8$. Following our problem formulation in Section 3, at each time step $t \in [T]$ (we set $T = 100$), each client $i \in [N]$ (we set $N = 20$) picks a data point $x_{t,i}$ from the candidate set $\mathcal{X}$, and then observes reward $y_{t,i}$ generated by function $f_1, f_2$ as mentioned above. Note that the values of both functions are negated, so by maximizing reward, the algorithms are trying to find data point that minimizes the function values. We should also note that the communication cost presented in the experiment results are defined as the total number of scalars transferred in the system (Wang et al., 2020), instead of number of time communication happens (Li & Wang, 2022a).

**Real-world dataset experiment setup & results**  To further evaluate Fed-GO-UCB's performance in a more challenging and practical scenario, we performed experiments using real-world datasets: MagicTelescope and Shuttle from the UCI Machine Learning Repository (Dua & Graff, 2017). We pre-processed these two datasets following the steps in prior works (Filippi et al., 2010), by partitioning the dataset in to 20 clusters, and using the centroid of each cluster as feature vector for the arm and its averaged response as mean reward. Then we simulated the federated bandit learning problem introduced in Section 3 with $T = 100$ and $N = 100$. From Figure 3, we can see that Fed-GO-UCB outperforms the baselines, with relatively low communication cost.

---

[1]We chose them from the test functions available in BoTorch package. See `https://botorch.org/api/test_functions.html` for more details.

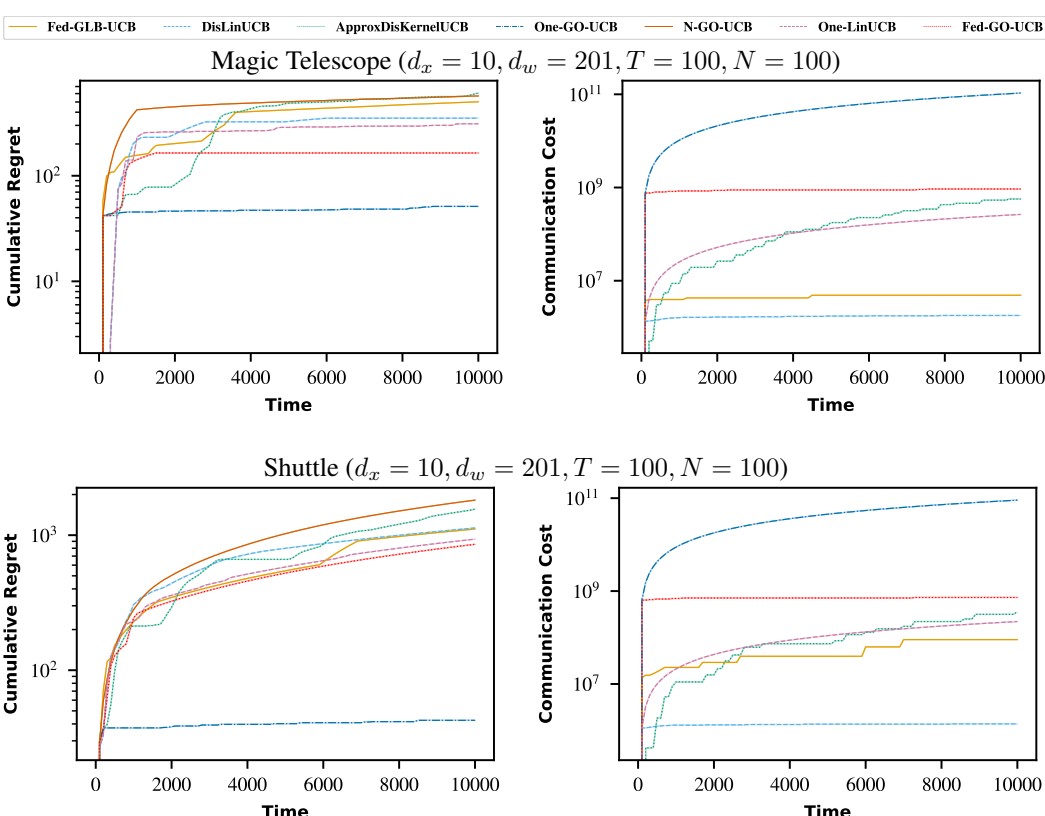

Figure 3: Comparison of cumulative regret and communication cost on real-world datasets.

