# OpenReview forum: "Communication-Efficient Federated Non-Linear Bandit Optimization"
_ICLR.cc/2024/Conference — ICLR 2024 poster_

### Official Review · Reviewer_6154 · 2023-10-31

**Soundness:** 3 good
**Presentation:** 2 fair
**Contribution:** 2 fair
**Rating:** 6
**Confidence:** 3

**Summary:**

The paper introduces Fed-GO-UCB, a federated bandit optimization algorithm designed for generic non-linear objective functions, addressing the limitations of existing methods that are confined to simplistic function classes. Federated optimization enables collaborative model estimation across decentralized datasets, ensuring data privacy and allowing large-scale computing. This is particularly beneficial for tasks requiring online interactions, such as next-word prediction in keyboard applications. Fed-GO-UCB operates under the coordination of a central server and multiple clients, ensuring data decentralization. The algorithm comprises two phases: uniform exploration and optimistic exploration, allowing clients to collaboratively minimize cumulative regret and make quality decisions during the learning process. The paper highlights the challenges in federated bandit optimization, particularly in constructing confidence sets for generic nonlinear functions and managing communication costs. Fed-GO-UCB addresses these issues through a novel confidence set construction and an efficient communication strategy. Empirical evaluations demonstrate the algorithm's superiority over existing federated bandit algorithms, particularly in approximating nonlinear functions. The paper also proves that Fed-GO-UCB achieves sub-linear rates for both cumulative regret and communication cost, making it a promising tool for decentralized machine learning applications involving sensitive data.

**Strengths:**

1. Novelty: Fed-GO-UCB is a new approach in federated bandit optimization for generic non-linear function optimization.

2. Theoretical Guarantees: The paper provides rigorous proofs for the sub-linear rates of cumulative regret and communication cost, ensuring the algorithm's reliability and efficiency.

3. Empirical Validation: The effectiveness of Fed-GO-UCB is demonstrated through extensive empirical evaluations, showcasing its superiority in approximating nonlinear functions and its practical applicability.

**Weaknesses:**

1. Limited Discussion on Assumptions: The paper mentions “some mild conditions” under which the algorithm performs well, but it could provide a more detailed discussion on these conditions and their practical implications.

2. Comparison with State-of-the-Art: While the paper demonstrates the superiority of Fed-GO-UCB over existing federated bandit algorithms, a more comprehensive comparison with state-of-the-art methods in decentralized machine learning would strengthen the paper's contributions.

3. Over-reliance on Communication: The need for occasional communications to aggregate local learning parameters may lead to potential inefficiencies or delays.

**Questions:**

1. How does the performance of Fed-GO-UCB compare with centralized global optimization methods, particularly in scenarios with a high number of clients?

2. Can Fed-GO-UCB be extended to handle heterogeneous clients with different reward functions, and if so, what modifications would be necessary?

3. What are the practical implications of the “mild conditions” under which Fed-GO-UCB operates, and how do these conditions influence the algorithm's applicability in real-world scenarios?

---

> ### Author Response · Authors · 2023-11-19
> **Response to Reviewer 6154 [Part 1/2]**
>
> **[Q1] Discussion on assumptions and their practical implications**
>
> Please find our answer to CQ1 in the common response for discussions about the assumptions made in our paper.
>
> Here we want to emphasize that the function class defined by Assumptions 1-3 generalizes the parametric function classes studied in existing federated bandit papers (Wang et. al. 2020; Li and Wang, 2022a; He et. al., 2022; Li and Wang, 2022b), and also covers additional nonlinear, nonconvex functions that have not been considered in existing works in federated bandits. We want to emphasize that this is already a nontrivial improvement.
>
> In terms of applicability in real-world scenarios, it is worth noting that linear and generalized linear models (both are special cases of ours) have already been shown to perform well in applications like news recommendation (Li et. al., 2010), and our generalization offers even more flexibility in the modeling choices during federated optimization, i.e., any parametric functions, including neural networks, that satisfy our local strongly convexity assumption.
>
> **[Q2] Comparison with decentralized machine learning methods**
>
> Thanks for the suggestion. We have added a section in Appendix B to discuss existing works in offline federated learning / decentralized machine learning to help further highlight the contribution of our paper.
>
> However, we want to clarify that, due to the fundamental differences in the problem formulation and learning objectives between federated bandit learning and decentralized machine learning, direct comparison in theoretical analysis and experiments is not possible.
> Specifically, the focus of decentralized machine learning is to collaboratively learn a good *point estimate* over a fixed dataset, i.e., convergence to the minimizer with fewer communications/iterations, while federated bandit learning requires collaborative *confidence set estimation* for efficient regret reduction over a finite time horizon $T$.
>
> One perspective to understand their relation is that federated bandit learning studies decision making over the whole time horizon $T$, while decentralized machine learning aims to solve the optimization problem on the dataset collected up to a fixed time step $t \in [T]$, i.e., the latter can be viewed as a component of the former. For example, in our Fed-GO-UCB algorithm, we can adopt any decentralized machine learning method as the *Oracle* to optimize Equation (2), but we still need the other components, e.g., confidence set construction (line 4) and OFU principle (line 3) for effective decision making. Therefore, these two are not directly comparable.
>
>
> **[Q3] The need for occasional communications may lead to inefficiencies or delays**
>
> We definitely agree with the reviewer that such communication “may lead to potential inefficiencies or delays”, but as discussed in our response to CQ2, in order to avoid the suboptimal regret $\tilde O(N \sqrt{T})$, communication among clients is necessary.
>
> This is why we need a well-designed federated bandit optimization method like Fed-GO-UCB to attain a good tradeoff between the conflicting objectives of regret minimization and communication efficiency, i.e., we wish to have the most regret reduction per communication. We have shown both theoretically and empirically, that our Fed-GO-UCB algorithm achieved this goal for the federated bandit optimization of nonlinear functions.

---

> > ### Author Response · Authors · 2023-11-19
> > **Response to Review 6154 [Part 2/2]**
> >
> > **[Q4] Comparison with centralized global optimization methods, in scenarios with a large number of clients**
> >
> > If I understand the question correctly, the reviewer is asking how small the regret of Fed-GO-UCB is when the number of clients is large, compared with centralized global optimization methods like GO-UCB (Liu and Wang, 2023).
> >
> > As discussed in Section 4.2, our Fed-GO-UCB algorithm attains the same $\tilde O(\sqrt{NT})$ regret upper bound as GO-UCB (note that as mentioned in our response to CQ2, $\tilde O(\sqrt{NT})$ is optimal as it matches the regret lower bound), while making sure only a sublinear communication cost is needed.
> >
> > More specifically, suppose there are $NT$ number of interactions/time steps in total. For Fed-GO-UCB, these $NT$ interactions are split to $N$ clients, each with $T$ interactions, and for GO-UCB, these $NT$ interactions are done on a centralized machine. The result mentioned in the previous paragraph is saying, Fed-GO-UCB can attain the same regret as GO-UCB, regardless of how many clients we are splitting the dataset to.
> >
> >
> > **[Q5] Necessary modification to handle heterogeneous clients**
> >
> >
> > Yes, Fed-GO-UCB can be extended to handle heterogeneous clients.
> >
> > Thanks to our confidence set construction, methods developed for federated linear bandits can now be readily applied on top of our framework to deal with nonlinear functions. Therefore, to handle heterogeneous clients, one can adopt the clustering methods proposed by Gentile et. al. (2014) and Li et. al. (2021) to cluster the clients into different groups (with clients sharing similar reward functions being in the same group), and then aggregate the statistics of clients in the same group for confidence set construction. Moreover, the communication protocol in our Algorithm 1 can still be used within each group, to reduce communication cost.
> >
> >
> > A recent work by Liu et. al. (2022) does exactly this for federated linear bandits, via a combination of the clustering method of Gentile et al. (2014) and the asynchronous communication protocol of Li and Wang (2022a). With the confidence sets constructed in our paper, a similar algorithm design as Liu et. al. (2022) can be used for heterogeneous clients with different nonlinear reward functions.
> >
> >
> > **Additional References**
> >
> > - Xutong Liu, Haoru Zhao, Tong Yu, Shuai Li, and John CS Lui. "Federated online clustering of bandits." In Uncertainty in Artificial Intelligence, pp. 1221-1231. PMLR, 2022.
> > - Claudio Gentile, Shuai Li, and Giovanni Zappella. "Online clustering of bandits." In International conference on machine learning, pp. 757-765. PMLR, 2014.
> > - Chuanhao Li, Qingyun Wu, and Hongning Wang. "Unifying clustered and non-stationary bandits." In International Conference on Artificial Intelligence and Statistics, pp. 1063-1071. PMLR, 2021.

---

> > ### Comment · Reviewer_6154 · 2023-11-21
> >
> > Thanks for the detailed response, solving my questions greatly. I will raise my score.

---

> > > ### Author Response · Authors · 2023-11-22
> > > **Response to Review 6154**
> > >
> > > We are glad to hear that we have addressed Reviewer 6154's questions regarding Assumption 1-3 and our contribution w.r.t. prior works, and we sincerely thank Reviewer 6154 again for the timely and constructive suggestions. Please let us know if you have any further questions.

---

### Official Review · Reviewer_yBxF · 2023-10-31

**Soundness:** 3 good
**Presentation:** 3 good
**Contribution:** 3 good
**Rating:** 6
**Confidence:** 4

**Summary:**

This paper studies the federated bandit optimization where the objective functions are non-linear yet i.i.d across $N$ agents. Existing works focus on either simplistic function classes or non-parametric function classes with bounded RKHS norms. In this paper, the authors propose a new Fed-GO-UCB algorithm, which contains two phases: the uniform exploration phase and the online learning phase. The authors prove that Fed-GO-UCB ahives $O(\sqrt{NT})$ regret while achieving $O(N^{1.5}\sqrt{T})$ communication cost. Finally, the authors conduct empirical experiments on both synthetic and real-world data to validate their theoretical results.

**Strengths:**

1. The problem setting is new and well-motivated. Compared with existing works that follow either simplistic function classes or non-parametric function classes with bounded RKHS norm, this work considers a more general non-linear form of objective function.
2. The results are sound and complete, with both theoretical analysis and empirical evaluation.
3. The algorithms and the analysis both have some novelty, for example, the two phase algorithms and the analysis built upon it.

**Weaknesses:**

1. Novelty: Though the setting is new and the two-phase design is interesting, after the uniform exploration (phase 1), it seems to me that one can combine the techniques from centralized non-linear bandit optimization problem with the communication protocol from the federated linear bandits (Li & Wang, 2022a) and federated generalized linear bandits (Li & Wang, 2022b).
2. Comparison with federated generalized linear bandits (Li & Wang, 2022b): From the algorithmic perspective, Li & Wang 2022b also uses some global updates (similar to uniform exploration) after the communication condition (line 6 of Algorithm 1) is satisfied, I am wondering if the current paper also uses this way to update $\hat{w}_0$, will there be any difference or improvement in the regret analysis? Moreover, I find that Li & Wang, 2022b can only achieve $O(dN^2\sqrt{T})$ communication. Since I suppose the current paper is more general and can cover Li & Wang 2022b, does this mean that the current paper can achieve a $O(\sqrt{N})$ improvement because of the difference in the algorithm design?
3. Lower bound: I do not find any discussion about the lower bound result or any discussion about it. Without it, one cannot see how tight the results are in terms of $$N, T$$ and all other parameters. I hope the authors can discuss the tightness of their results during the rebuttal.

**Questions:**

Please justify or comment on the three weaknesses above.

---

> ### Author Response · Authors · 2023-11-19
> **Response to Reviewer yBxF**
>
> **[Q1] Novelty**
>
> As mentioned in the Technical novelties paragraph in Section 1, the approximation method from centralized non-linear bandit optimization (Liu and Wang, 2023) cannot be applied to the federated setting. Specifically, the confidence sets for the optimal parameter $\omega^\star$ constructed in (Liu and Wang, 2023) is $\{\omega: \lVert \omega - \hat \omega_t \rVert_{\Sigma_{t}} \leq \beta_{t}\}$, where $\Sigma_t =\lambda \mathbf{I} + \sum_{s=1}^{t} \nabla f_{x_s}(\hat \omega_{s}) \nabla f_{ x_s}(\hat \omega_{s})^\top$ and $b_t =\sum_{s=1}^{t} \nabla f_{x_s}(\hat \omega_{s}) \bigl[\nabla f_{ x_s}(\hat \omega_{s})^\top \hat \omega_{s} + y_s - f_{x_s}(\hat \omega_{s}) \bigr]$, where the gradients $\nabla f_{x_s}(\hat \omega_{s})$ have to be calculated w.r.t. the model $\hat \omega_{s}$ at step $s$.
>
> Due to local model updates in federated settings, now each client $i$ has a different sequence of local models $\hat \omega_{s,i}$ for $s=1,2,\dots,t$. By directly aggregating the statistics $\Sigma_{t,i}$ and $b_{t,i}$ computed based on such $N$ different sequences of local models, one cannot obtain a valid confidence set. To address this issue, we improve the analysis of Liu and Wang (2023), and show that we can construct a valid confidence set using the gradients w.r.t. a common initial model $\hat \omega_0$, instead of clients’ locally updated ones.
>
> Therefore, after obtaining $\hat \omega_0$ at the end of Phase I, the clients can collaborate on confidence set estimation via a common embedding function defined by the gradient $\nabla f_{x_s}(\hat \omega_{0})$. This is why we are able to utilize the communication protocol designed for federated linear bandits (Wang et. al. 2020). We want to emphasize that being able to utilize communication protocols designed for federated linear bandits for nonlinear functions via our novel confidence set construction is already non-trivial and may be of independent interest for future research. For example, one can also extend our method to asynchronous protocols (Li and Wang, 2022a; He et. al, 2022), as well as heterogeneous clients setting (Liu et. al. 2022). Moreover, as discussed in our response to CQ2, this new algorithm design helps us improve upon FedGLB-UCB (Li and Wang, 2022b) by a factor of $d_x \sqrt{N}$ for federated optimization of nonlinear functions, because in Phase II we are able to use the efficient closed-form update as federated linear bandits, instead of the expensive iterative optimization as FedGLB-UCB.
>
>
> **[Q2] Improvement compared with federated generalized linear bandits (Li & Wang, 2022b)**
>
> Yes, our paper has an $\tilde O(d_x \sqrt{N})$ improvement on communication cost thanks to the new algorithm design. Please see our answer to CQ2 in the general response for more details about how this is achieved.
>
> **[Q3] Tightness of communication cost upper bound**
>
> Please see our answer to CQ2 in the common response for discussions on the tightness of our communication cost upper bound in terms of $N$ and $T$.
>
>
> **Additional References**
>
> - Xutong Liu, Haoru Zhao, Tong Yu, Shuai Li, and John CS Lui. "Federated online clustering of bandits." In Uncertainty in Artificial Intelligence, pp. 1221-1231. PMLR, 2022.

---

> > ### Author Response · Authors · 2023-11-22
> > **Response to Reviewer yBxF**
> >
> > We thank Reviewer yBxF for the constructive comments. We would like to follow up with Reviewer yBxF regarding our responses to the questions about our technical novelty, improvement over FedGLB-UCB and tightness of our communication upper bound, and understand if all your concerns have been properly addressed, given the author rebuttal period is coming to its end.
> >
> > Please let us know if you have any further questions, and we look forward to the possibility of your updated evaluation of our work.

---

### Official Review · Reviewer_emcJ · 2023-11-01

**Soundness:** 3 good
**Presentation:** 3 good
**Contribution:** 2 fair
**Rating:** 6
**Confidence:** 2

**Summary:**

This paper presents a communication-efficient federated algorithm, Fed-GO-UCB, for a bandit optimization problem with non-linear function. Their analysis shows that the regret upper bound of Fed-GO-UCB matches that of Fed-GO-UCB's centralized counterpart with sub-linear communication costs. The authors explain the main logics behind analysis in details. Empirical experiment results are also included.

**Strengths:**

- This work is one of the pioneering effort in studying federated bandit optimization with non-linear function
- This work generalizes a distributed regression theoretical guarantee to account for approximation error, which may be of interest to the federated learning community
- This paper is in general well written and easy to follow

**Weaknesses:**

- The literature review provided is somewhat brief and limited in scope and detail
- Though this work features considering non-linear function optimization, it seems that its main contribution is on addressing the challenges of federated setting. Could the authors discuss more about the special challenges of non-linear function optimization?

**Questions:**

- The novel technical contributions of this work are Lemma 6 and 8 if I understand correctly. Shouldn't these two lemmas be named as Propositions instead?
- Could the authors comment on the difficulties in obtaining Lemma 6 and 8? Which part of difficulties is due to the federated setting and which part of the difficulties is due to the non-linear function?
- Could the authors comment on the tightness of the communication cost bound?

---

> ### Author Response · Authors · 2023-11-19
> **Response to Reviewer emcJ**
>
> **[Q1] Additional discussions on related works**
>
> We agree with Reviewer emcJ that more detailed and comprehensive discussions on related works would better highlight the contribution of our paper, so in Appendix B we have added a section to discuss related works in federated bandit learning and offline federated learning, as well as a table summarizing existing federated bandit algorithms. Please also see our answer to CQ2 in the common response for a detailed discussion about theoretical comparison with existing works on federated bandit learning.
>
>
> **[Q2] Challenges of nonlinear function optimization**
>
> We want to clarify that the unique challenge addressed in this paper is bandit optimization of **nonlinear functions under federated settings**. We cannot view them separately as suggested by the reviewer. Specifically, federated setting requires communication efficiency, and bandit optimization of nonlinear functions requires construction of confidence sets. Therefore, the unique question answered in our paper is, how to construct confidence sets for the nonlinear functions collaboratively by all clients in a communication efficient manner, which cannot be addressed by prior works.
>
>
> As mentioned in the Technical novelties paragraph in Section 1 of our paper, the approximation method we propose is a non-trivial extension of (Liu and Wang, 2023), i.e., their method cannot be applied to federated settings as their approximation relies on a sequence of continuously updated models $\hat \omega_{s}$ for $s=1,2,\dots,t$. But in federated settings, each client has a different sequence of locally updated models, which leads to difficulty in aggregating local statistics (see also our answer to Q1 of Reviewer yBxF). This problem does not exist for federated linear bandits, since the confidence set for linear models can be directly constructed using feature vector $x$, instead of gradient $\nabla f_{ x_s}(\hat{\omega}_{s})$.
>
>
> To address this issue, we propose a new approximation procedure with improved analysis over Liu and Wang (2023), such that the statistics computed by all clients are now based on a common embedding function $\nabla f_{x_s}(\hat{\omega}_{0})$, i.e., the gradient w.r.t. the shared model $\hat \omega_0$. Note that $\hat \omega_0$ is obtained at the end of Phase I and then always remains fixed.
>
>
> Compared with prior works in federated linear bandits, which conveniently enjoys closed-form solutions, nonlinear function faces another challenge, i.e., it requires iterative optimization for each model update, which is expensive in federated settings. As mentioned in our answer to CQ2, prior work that studies generalized linear models [Li and Wang 2022b] needs to call the distributed regression oracle during each global synchronization to execute such an expensive iterative optimization procedure. We alleviate this issue by only executing the iterative optimization at the end of Phase I, and then resorting to approximated function value during Phase II, so that more efficient communication is enabled thanks to its closed-form update.
>
> **[Q3] Lemma 6 and Lemma 8**
>
>
> Yes, Lemma 6 and Lemma 8 are the main technical contributions of this paper. We currently name them as Lemmas instead of Propositions, because they will be invoked as intermediate steps to prove Theorem 5, instead of being standalone technical results.
>
>
> As mentioned in Section 5.1, prior work in centralized nonlinear bandit optimization (Liu and Wang, 2023) assumes the distributed regression oracle outputs the exact minimizer of Equation (2). However, this is unreasonable in a federated setting, since finding the exact minimizer would require an infinite number of iterations, which is not communication efficient. Therefore, when deriving Lemma 6, we need to take into account the additional approximation error $\epsilon$ in optimizing Equation (2), and analyze the number of iterations required to attain $\epsilon$ (this is needed in the communication cost analysis later).
>
>
> For the difficulty in constructing the confidence sets in Lemma 8, please see our response to Q2.
>
>
> **[Q4] Tightness of the communication cost upper bound**
>
> Please see CQ2 in the common response for discussions on tightness of the communication cost bound.

---

> ### Comment · Reviewer_emcJ · 2023-11-22
>
> I appreciate the authors' response to my questions and the revisions in their paper.

---

> > ### Author Response · Authors · 2023-11-22
> > **Response to Review emcJ**
> >
> > We want to thank Reviewer emcJ again for the appreciation of our work, and constructive suggestions on literature review as well as discussions about tightness of our communication upper bound! Please let us know if you have any further questions, and we look forward to the possibility of your updated evaluation of our work.

---

### Official Review · Reviewer_rFun · 2023-11-02

**Soundness:** 3 good
**Presentation:** 3 good
**Contribution:** 3 good
**Rating:** 6
**Confidence:** 3

**Summary:**

This paper's primary focus is on tackling the non-linear bandit optimization problem within a federated setting. It introduces the innovative Fed-GO-UCB algorithm, which represents a substantial improvement over previous centralized algorithms. Remarkably, this federated algorithm achieves a theoretical regret guarantee comparable to previous centralized approaches, specifically $O(\sqrt{NT})$, while also demonstrating sub-linear communication complexity. Additionally, the empirical results robustly confirm the efficiency and effectiveness of the proposed algorithm.

**Strengths:**

1. The proposed algorithm excels in providing a near-optimal guarantee for federated learning with generic non-linear function optimization.

2. The empirical results robustly confirm the efficiency and effectiveness of the proposed algorithm.

3. This paper is well-written and easily comprehensible.

**Weaknesses:**

1. The theoretical guarantee of Fed-GO-UCB relies on several critical assumptions (Assumptions 1 to 3), and it may be challenging to establish whether these assumptions hold in common situations, such as neural networks with ReLU activation functions. Even if the assumptions hold, determining the values of the parameters in Fed-GO-UCB that depend on these assumptions can be a non-trivial task. It remains an open question how to calculate these constants and set the algorithm's parameters effectively in practical applications.

2. The communication complexity, as indicated by Theorem 5, is stated as $O(\sqrt{T})$, which is significantly higher than the communication complexity of $O(\log T)$ achieved by previous works. Presenting this level of communication cost as efficient without providing lower bound results for communication complexity can be misleading. Furthermore, the experimental results in Figure 2 show a communication complexity of $10^7$ for round $T=500$, which seems highly inefficient in practice.

3. In the context of federated learning, there is often a tradeoff between communication complexity and performance, specifically in terms of regret guarantee. It's a common observation that algorithms with higher communication complexity can potentially achieve better performance. To gain a clearer understanding of the algorithms' relative performances, it would be beneficial for the author to conduct experiments where the selected parameters makes different algorithms have similar communication complexity. This approach would enable a direct and fair comparison that isolates the impact of communication complexity from the approximation capabilities, providing more conclusive insights into the algorithm's efficiency and effectiveness in a practical federated learning setting.

**Questions:**

1. In the Fed-GO-UCB algorithm, it's important to clarify that agents should reset $\Delta\Sigma$ and $\Delta b$ to zero after uploading their respective datasets.

2. In Figure 3, the observation that the communication complexity for several algorithms does not start from zero at time $T=0$ can be puzzling. It would be beneficial if the author could offer an explanation for this behavior.

3. For the Theorem 5, is it possible to provide a corresponding lower bound to suggest that $\Omega(\sqrt{T})$ communication is necessary. For instance, Min el al., 2023 [1] provide a lower bound showing
that a minimal $\Omega(dM)$ communication complexity is required to improve the performance of linear bandit (or MDPs) through
collaboration. Insights into lower communication complexity bounds would strengthen the support for the Fed-GO-UCB algorithm and provide a more comprehensive understanding

[1] Cooperative Multi-Agent Reinforcement Learning: Asynchronous Communication and Linear Function Approximation

---

> ### Author Response · Authors · 2023-11-19
> **Response to Reviewer rFun [Part 1/2]**
>
> **[Q1] Whether the assumptions hold in common situations and how to determine the values of the parameters**
>
> Please find our answer to CQ1 in the common response for discussions about the assumptions made in our paper.
>
> Here we want to emphasize that, though still with its limitations, the function class defined by Assumptions 1-3 generalizes the parametric function classes studied in existing federated bandit papers (Wang et. al. 2020; Li and Wang, 2022a; He et. al., 2022; Li and Wang, 2022b), and also covers additional nonlinear, nonconvex functions that have not been considered in prior works. It is worth noting that linear and generalized linear models (both are special cases of ours) have already been shown to perform well in real-world applications like news recommendation (Li et. al., 2010), and our generalization offers even more flexibility in the modeling choices during federated optimization.
>
> Moreover, we want to clarify that, Fed-GO-UCB algorithm does not need to know the values of the constants appearing in Assumptions 1-3, except for the bound of function value $F$ and the strong convexity parameter $\mu$ in order to set the width of the confidence set $\beta$ (as given in Lemma 8). Both constants are assumed to be known in bandit learning literature, e.g. linear bandits (Abbasi-yadkori, 2011; Wang et. al., 2020), logistic bandits (Faury et. al., 2020), and generalized linear bandits (Filippi et. al., 2010; Li and Wang, 2022b), so we are not introducing any additional assumption on the knowledge of these constants compared with prior works.
>
> In practical applications, since these two constants, and the standard deviation $\sigma$ of reward noise that also appears in $\beta$, are unknown before observing the data, people typically directly tune the value of $\beta$ when applying bandit algorithms under different scenarios. But we agree that it would be an interesting future direction to investigate whether one can design no-regret bandit algorithms that do not require such knowledge as input, i.e., automatically adapt the strength of exploration to these unknown constants in an online manner.
>
>
> **[Q2] Tightness of communication cost upper bound and comparison with prior works**
>
> Please find our answer to CQ2 in the common response about the communication lower bound and detailed comparison with prior works.
>
> Specifically, we want to clarify that, the mentioned $O(\log(T))$ rate is attained by prior works on federated linear bandits (Wang et. al., 2020; Li and Wang, 2022a; He et al., 2022; Min et al. 2023), and federated kernelized bandits (Li et al., 2022; Li et al., 2023), where closed-form solutions are available, so that only one round of communication is needed to aggregate the local sufficient statistics for model update.
>
> In comparison, our work is more similar to (Li and Wang, 2022b) that studies federated generalized linear bandits, in the sense that, both consider nonlinear functions with no closed-form solution, and thus iterative optimization is needed to compute model update. This is intrinsically more expensive, because at least $\Omega(\sqrt{NT})$ rounds of communication is needed to collect and aggregate local gradients/models when calling the distributed regression oracle (see the discussions in Section 4.1 and 4.2 of Li and Wang (2022b), as well as the lower bound result of Arjevani and Shamir (2015)). Moreover, as discussed in our answer to CQ2, we strictly improve upon the result of Li and Wang (2022b) by a factor of $d_x \sqrt{N}$, since we only need to call the regression oracle once at the end of Phase I, instead of at each global synchronization as Li and Wang (2022b). As shown in the table in CQ2, Fed-GO-UCB is currently the most communication efficient solution for federated bandit optimization of nonlinear function, which is already a nontrivial contribution to the literature.

---

> > ### Author Response · Authors · 2023-11-19
> > **Response to Reviewer rFun [Part 2/2]**
> >
> > **[Q3] Empirical comparison of communication efficiency**
> >
> > Here we want to clarify a potential misunderstanding in the `fair comparison’ suggested by Reviewer rFun.
> > We definitely agree that higher communication cost leads to better (at least no worse) performance, but this monotonicity is only true when we fix the function class unchanged. Comparison across different function classes introduce additional intricacies.
> >
> > Intuitively, a function that involves more parameters requires more samples to estimate, e.g., an overparameterized neural network vs a linear regression model. Therefore, a federated bandit algorithm using the overparameterized neural network may perform worse than that with a simple linear model, when there is not enough communication, or when the dataset is `easy’, i.e., linear function already fits it pretty well. However, this does not mean the former is a worse algorithm than the latter.
> >
> > The proper choice of function class depends on the dataset (i.e., the underlying reward mapping), as well as the communication bandwidth of the application at hand. The advantage of our Fed-GO-UCB over prior works, e.g., DisLinUCB and FedGLB-UCB, is that **it offers a larger variety of modeling choices**, so that it would be easier to find a suitable function class for different application scenarios. In general, for datasets that can be well-fitted by a linear model, DisLinUCB enjoys the advantage of its closed-form model updates, which is much more communication-efficient than gradient-based updates. However, in the more common situations where linear models give poor performance and one has to employ more complicated models at the cost of communication efficiency (because closed-form solution no longer exists), DisLinUCB no longer applies, while Fed-GO-UCB addresses this in a provably efficient manner.
> >
> > To justify this argument, we have **conducted additional experiments that add One-LinUCB** (DisLinUCB algorithm with communication threshold set to $-\infty$, so that communication happens in each time step) into comparison. Please see the updated Figure 2 and Figure 3 in our paper for more details. From the experiment results, we can see that, though with maximum communication possible, One-LinUCB is still outperformed by Fed-GO-UCB by a large margin in terms of cumulative regret, which clearly shows the limited capability of linear models in fitting the dataset. In this case, more complex models are preferred, and our Fed-GO-UCB algorithm offers a principled way to balance regret and communication.
> >
> > **[Q4] Reset $\Delta \Sigma$ and $\Delta b$ to zero after uploading**
> >
> > Thanks for the suggestion. We have updated the description of Algorithm 1 to clarify this.
> >
> >
> > **[Q5] Communication cost does not start from zero in experiment results**
> >
> >
> > Apologize for the confusion. This is because the point $(0,0)$ was not plotted. We have updated the figures, so now the communication cost of all algorithms start from $0$.
> >
> >
> > The line plot in Figure 3 is made by connecting 100 points, where each point corresponds to the communication cost at the end of each round (there are $T=100$ rounds in total, and in each round $N=100$ clients interact with the environment in a round-robin manner, so in total there are $NT=10000$ time steps). The observation that the communication cost of the first point is larger than 0 is simply because communications have happened during the first $100$ interactions in round $T=1$.
> >
> >
> > **Additional Reference**
> >
> > - Yossi Arjevani, and Ohad Shamir. "Communication complexity of distributed convex learning and optimization." Advances in neural information processing systems 28 (2015).
> > - Louis Faury, Marc Abeille, Clément Calauzènes, and Olivier Fercoq. "Improved optimistic algorithms for logistic bandits." In International Conference on Machine Learning, pp. 3052-3060. PMLR, 2020.

---

> > > ### Comment · Reviewer_rFun · 2023-11-21
> > >
> > > Thanks for the response and I will keep my positive score.

---

> > > > ### Author Response · Authors · 2023-11-22
> > > > **Response to Reviewer rFun**
> > > >
> > > > We want to thank Reviewer rFun once again for the valuable feedback. We have carefully considered your comments and suggestions on Assumption 1-3 and communication complexity compared with existing works, and we believe we have adequately addressed the concerns you raised. If you have any further questions or suggestions, please do not hesitate to let us know.

---

### Author Response · Authors · 2023-11-19
**Common Response to All Reviewers [Part 1/3]**

We sincerely thank all the reviewers for their thoughtful comments and constructive suggestions, which would significantly help us strengthen our paper. Based on the reviewers' comments, we have added additional discussions about related works and a table comparing the theoretical results of existing federated bandit algorithms in Appendix B, as well as updating Figure 2 and 3 with new experiment results.

It is worth noting that there are some shared comments regarding *the assumptions made in the paper* and *tightness of our communication upper bound*. We now first provide our answers to these common questions, and endeavor to provide individual responses to each reviewer.
We are fully committed to engaging with the discussions, if any further information or clarification is needed regarding our response.

**[CQ1] Assumptions Made in our Paper (Reviewer rFun, 6154)**

The realizability assumption (Assumption 1), in essence, allows us to ground our theoretical framework in a setting where the unknown objective function $f$ exists within the specified parametric function class, so that there is no additional regret caused by misspecification.
This is a very weak assumption that is commonly made, either explicitly or implicitly, in bandit learning literature (Wei et. al., 2011; Foster and Rakhlin, 2020; Foster et. al., 2018). Similarly, bounded, differentiable and smooth function (Assumption 2) is also commonly assumed in bandit learning literature, as well as (online) convex optimization in general (Hazan, 2016).

Note that Assumption 3 is made on the *expected loss* $L(\omega)$ over the uniform exploration distribution (not on the unknown objective function $f$ itself), and it is only needed in Phase I to establish the convergence result of $\lVert \hat{\omega}_{0}-\omega^\star \rVert_2 = O(1/\sqrt{T_0 })$.
We want to clarify that, though this assumption may look unfamiliar at first glance, it is *strictly weaker than the global strongly convex expected loss in prior works studying parametric bandits*, e.g., linear and generalized linear bandits (Filippi et. al., 2010), because it does not require convexity except in a local region near the optimal parameter $\omega^\star$, i.e., $\{\omega: \lVert \omega - \omega^\star \rVert_2 \leq (\tau/\mu)^\frac{1}{2-\gamma}\}$. This allows for an arbitrary number of spurious local minima in the landscape of the expected loss function, thus covering unknown objective functions that can be highly non-convex and non-linear. We want to emphasize that this is already a nontrivial improvement over existing works in federated bandits.

There is also a line of works studying bandit problems with neural function approximation (Zhou et al., 2020; Zhang et al., 2020; Dai et al., 2022), but they are still intrinsically linear due to the kernelized bandit analysis (with a Neural Tangent Kernel), and thus only applies when the width of the neural network is extremely large, e.g., $\tilde O (T^4 |\mathcal X|^4)$. In contrast, our results apply to generic nonlinear function approximations and do not need such overparameterization.

---

> ### Author Response · Authors · 2023-11-19
> **Common Response to All Reviewers [Part 2/3]**
>
> **[CQ2] Tightness of Communication Cost Upper bound (Reviewer rFun, emcJ, yBxF)**
>
> We agree with the reviewers that discussions about the tightness of communication cost upper bound can provide more insights into the efficiency of Fed-GO-UCB algorithm. In the following paragraphs, we will discuss the known communication lower bound result, as well as comparison with existing works in detail.
>
> First, we should note that, for federated bandit optimization, comparison on communication cost is only meaningful when fixing certain constraints on regret. The following two extreme cases are of particular interests:
>
> 1. run $N$ instances of an optimal bandit algorithm, e.g. LinUCB (Abbasi-yadkori et. al. 2011) for linear functions and GO-UCB (Liu and Wang, 2023) for generic nonlinear functions in our problem, separately on each client with no communication, which leads to $\tilde O(N\sqrt{T})$ regret and $0$ communication cost;
>
> 2. run one instance of the optimal bandit algorithm over the data of all $N$ clients, which leads to $\tilde O(\sqrt{NT})$ regret and communication cost $O(N^2 T)$.
>
> The regret upper bound in case 2 is optimal, since it already matches the regret lower bound for a centralized bandit problem with $NT$ interactions. Therefore, the typical goal of federated bandit optimization is to attain the optimal regret of $\tilde O(\sqrt{NT})$, while having a communication cost sub-linear in $T$ (this is what “efficient communication” means).
>
> The main contribution of our paper is to propose the first algorithm that achieves such a goal for federated bandit optimization with generic nonlinear objective functions. The lower bound analysis for the communication cost of federated bandits is highly non-trivial, and it still remains an open problem to close the gap between the communication upper bound attained by existing works and the known lower bound.
>
> To the best of our knowledge, the only communication lower bound result available states that, in order to have $o(N\sqrt{T})$ regret, an $\Omega(N)$ communication cost is necessary (Wang et. al., 2020; He et. al., 2022; Min et. al., 2023). It was originally derived for the context-free bandits (Theorem 2 of (Wang et. al., 2020)), but as shown in Theorem 5.3 of (He et. al., 2022), it also applies to federated linear bandits, and thus applies to our problem as well. And as mentioned by Reviewer rFun, Min et. al. (2023) further applied this analysis to linear MDPs.
> However, none of these existing works can close the gap with the $\Omega(N)$ communication lower bound. Specifically, to obtain the optimal $\tilde O(\sqrt{NT})$ regret, DisLinUCB (Wang et. al., 2020) requires $\tilde O(N^{1.5})$ communication cost,  and FedLinUCB (He et. al., 2022) requires $\tilde O(N^{2})$ communication cost, with the former one having the same rate in $N$ as our Fed-GO-UCB. This indicates that, **in terms of dependence on $N$, our result is already on par with the state of the art**. It is an important future direction to this line of research to further close this gap, e.g., with a tighter lower bound analysis showing the necessary amount of communication for the optimal $\tilde O(\sqrt{NT})$ regret.

---

> ### Author Response · Authors · 2023-11-19
> **Common Response to All Reviewers [Part 3/3]**
>
> **Continue to answer CQ2**
>
> Apart from Wang et. al. (2020), He et. al. (2022), and Min et. al. (2023) who study linear models, where only one round of communication is required to compute each closed-form model update via aggregating local sufficient statistics, FedGLB-UCB (Li and Wang, 2022b) considers nonlinear models (generalized linear models) for federated bandits, where closed-form solutions do not exist and thus iterative optimization is required, which is similar to the challenge we are facing. It is worth noting that their communication cost $\tilde O(N^2 \sqrt{T})$ also has a $\sqrt{T}$ term caused by the iterative optimization procedure of the regression oracle (i.e., it requires $O(\sqrt{NT})$ iterations of gradient aggregations), which is similar to our Fed-GO-UCB algorithm. Moreover, as pointed out in their Section 4.2, as long as iterative optimization is used to compute model update, the $O(\sqrt{NT})$ factor is unavoidable (otherwise the obtained model is too far away from the minimizer of the loss function).
>
> Though we cannot further reduce the number of iterations required when calling the regression oracle, we still managed to **improve upon the communication cost of FedGLB-UCB by a factor of $\tilde O(d_x \sqrt{N})$**, since we only need to call the regression oracle once at the end of Phase I, instead of calling the oracle during each global synchronization as FedGLB-UCB (and there are $\tilde O(d_x \sqrt{N})$ global synchronizations in total). That being said, due to the lack of tighter lower bound analysis for federated bandits as we discussed earlier, it is still not clear whether the $\sqrt{T}$ dependence can be further reduced for nonlinear functions. However, we argue that it is already a nontrivial contribution to strictly improve the communication cost of FedGLB-UCB for federated optimization of nonlinear functions. Moreover, it is also safe to hypothesize that, unless one can completely remove the need of iterative optimization for nonlinear functions, our Fed-GO-UCB algorithm already attains the optimal rate as it only requires calling the distributed regression oracle once.
>
> In the following table, we have summarized the theoretical guarantees of the aforementioned works in federated bandits. Hopefully this would provide a clearer picture of our contribution.
>
> | Related Works        | Modeling Assumption | Protocol     | Regret          | Communication     |
> |----------------------|---------------------|--------------|-----------------|-------------------|
> | (Wang et. al., 2020) | linear              | synchronous  | $\sqrt{NT}$     | $N^{1.5}$         |
> | (Li and Wang, 2022a) | linear              | asynchronous | $\sqrt{NT}$     | $N^{2}$           |
> | (He et. al., 2022)   | linear              | asynchronous | $\sqrt{NT} + N$ | $N^{2}$           |
> | (Li and Wang, 2022b) | generalized linear  | synchronous  | $\sqrt{NT}$     | $N^{2}\sqrt{T}$   |
> | (Li et. al., 2022)   | kernel              | synchronous  | $\sqrt{NT}$     | $N^{2}$           |
> | (Li et. al., 2023)   | kernel              | asynchronous | $\sqrt{NT} + N$ | $N^{2}$           |
> | (Dai, et. al., 2022) | kernel (NTK)        | synchronous  | $ N\sqrt{T}$    | $N^{1.5}T^{2}$    |
> | Ours                 | nonconvex           | synchronous  | $\sqrt{NT}$     | $N^{1.5}\sqrt{T}$ |
>
>
> **Additional References**
>
> - Wei Chu, Lihong Li, Lev Reyzin, and Robert Schapire. "Contextual bandits with linear payoff functions." In Proceedings of the Fourteenth International Conference on Artificial Intelligence and Statistics, pp. 208-214. JMLR Workshop and Conference Proceedings, 2011.
> - Dylan Foster, Alekh Agarwal, Miroslav Dudík, Haipeng Luo, and Robert Schapire. "Practical contextual bandits with regression oracles." In International Conference on Machine Learning, pp. 1539-1548. PMLR, 2018.
> - Elad Hazan. "Introduction to online convex optimization." Foundations and Trends® in Optimization 2, no. 3-4 (2016): 157-325.
> - Dongruo Zhou, Lihong Li, and Quanquan Gu. "Neural contextual bandits with ucb-based exploration." In International Conference on Machine Learning, pp. 11492-11502. PMLR, 2020.
> - Weitong Zhang, Dongruo Zhou, Lihong Li, and Quanquan Gu. "Neural thompson sampling." arXiv preprint arXiv:2010.00827, 2020.
> - Yifei Min, Jiafan He, Tianhao Wang, and Quanquan Gu. "Multi-agent Reinforcement Learning: Asynchronous Communication and Linear Function Approximation." arXiv preprint arXiv:2305.06446, 2023.

---

> > ### Comment · Area_Chair_f3Eh · 2023-11-20
> > **Please respond to authors' comments**
> >
> > Dear reviewers,
> >
> > The authors have provided detailed replies to your comments. Please go through them and at least acknowledge that you've read through them regardless of whether or not you change your score.
> >
> > Regards,
> >
> > Your AC

---

### Meta-Review · Area_Chair_f3Eh · 2023-12-05

**Metareview:**

This paper studies a rather general problem setting of federated non-linear bandit optimization. This greatly enhances the function classes that were considered previously. By devising and analyzing a two-phase procedure, the authors proved an achieveable regret bound that is sublinear in T. The communication rate is also small (sublinear). These contributions alone justify the acceptance of the paper, even though the lower bound, which I agree is challenging to derive in this setting, is not available.

**Justification For Why Not Higher Score:**

The work could benefit from a lower bound, though I agree that it is challenging to derive.

**Justification For Why Not Lower Score:**

The setting is novel and the contributions are rather evident. There is unanimity among the reviewers in their positive view of this paper.

---

### Decision · Program_Chairs · 2024-01-16

Accept (poster)